# Influence of aromatics on tropospheric gas-phase composition

Domenico Taraborrelli[1], David Cabrera-Perez[2], Sara Bacer[2,*], Sergey Gromov[2], Jos Lelieveld[2], Rolf Sander[2], and Andrea Pozzer[2,3]

[1]Institute of Energy and Climate Research (IEK-8), Forschungszentrum Jülich GmbH, 52425 Jülich, Germany
[2]Atmospheric Chemistry Department, Max-Planck Institute of Chemistry, Hahn-Meitner-Weg 1, 55128 Mainz, Germany
[3]International Centre for Theoretical Physics, 34100 Trieste, Italy
[*]Now at: Université Grenoble Alpes, CNRS, Grenoble INP, LEGI, 38000 Grenoble, France

**Correspondence:** D. Taraborrelli (d.taraborrelli@fz-juelich.de)

**Abstract.**

Aromatics contribute a significant fraction to organic compounds in the troposphere and are mainly emitted by anthropogenic activities and biomass burning. Their oxidation in lab experiments is known to lead to the formation of ozone and aerosol precursors. However, their overall impact on tropospheric composition is uncertain as it depends on transport, multiphase chemistry, and removal processes of the oxidation intermediates. Representation of aromatics in global atmospheric models has been either neglected or highly simplified. Here, we present an assessment of their impact on the gas-phase chemistry, using the general circulation model EMAC (ECHAM5/MESSy Atmospheric Chemistry). We employ a comprehensive kinetic model to represent the oxidation of the following monocyclic aromatics: benzene, toluene, xylenes, phenol, styrene, ethylbenzene, trimethylbenzenes, benzaldehyde, and lumped higher aromatics that contain more than 9 C atoms.

Significant regional changes are identified for several species. For instance, glyoxal increases by 130 % in Europe and 260 % in East Asia, respectively. Large increases in HCHO are also predicted in these regions. In general, the influence of aromatics is particularly evident in areas with high concentrations of $NO_x$, with increases up to 12 % in $O_3$ and 17 % in OH.

On a global scale, the estimated net changes of trace gas levels are minor when aromatic compounds are included in our model. For instance, the tropospheric burden of CO increases by about 6 %, while the burdens of OH, $O_3$, and $NO_x$ (NO + $NO_2$) decrease between 3 % and 9 %. The global mean changes are small, partially because of compensating effects between high- and low-$NO_x$ regions. The largest change is predicted for the important aerosol precursor glyoxal, which increases globally by 36 %. In contrast to other studies, the net change in tropospheric ozone is predicted to be negative, -3 % globally. This change is larger in the northern hemisphere where global models usually show positive biases. We find that the reaction with phenoxy radicals is a significant loss for ozone, of the order of 200-300 Tg/yr, which is similar to the estimated ozone loss due to bromine chemistry.

Although the net global impact of aromatics is limited, our results indicate that aromatics can strongly influence tropospheric chemistry on a regional scale, most significantly in East Asia. An analysis of the main model uncertainties related to oxidation and emissions suggests that the impact of aromatics may even be significantly larger.

# 1 Introduction

Volatile organic compounds (VOCs) comprise a large variety of species which influence the tropospheric chemistry at local, regional, and global scales. VOCs react mainly with the hydroxyl radical (OH), ozone ($O_3$), and the nitrate radical ($NO_3$), or they are photolyzed. Their oxidation affects many key atmospheric species, including OH, $O_3$, and nitrogen oxides ($NO_x$ = NO + $NO_2$). Production and destruction of ozone is controlled by the ratio of VOCs to $NO_x$. In the low-$NO_x$ regime, the net effect of VOC oxidation is ozone destruction. Under high-$NO_x$ conditions, e.g., in urban areas, $O_3$ is generated by the

oxidation of VOCs (Sillman et al., 1990).

Aromatics are unsaturated planar cyclic organic compounds with enhanced stability due to a strong electron delocalization. Several of them are present in the atmosphere, e.g., benzene, toluene, ethylbenzene, xylenes, styrene and trimethylbenzenes. In general, aromatic compounds are found in continental areas, especially in industrialized urban and semi-urban regions (Barletta et al., 2005) where their emissions are highest. They are responsible for a considerable fraction of ozone and secondary organic

aerosol (SOA) formation (Ng et al., 2007; Lee et al., 2002; Ran et al., 2009). In addition, many aromatics are toxic (WMO, 2000).

Emissions of aromatics are primarily anthropogenic, related to fuel combustion, and leakage from fuels and solvents (Koppmann, 2007; Sack et al., 1992). Emissions from biomass burning play a secondary role but can be important on a regional scale (Cabrera-Perez et al., 2016). Biogenic emissions are only relevant for toluene (Heiden et al., 1999), although recent studies

suggest that other aromatics from biogenic sources may rival those from fossil fuel use (Misztal et al., 2015).

As shown by Cabrera-Perez et al. (2016), aromatic compounds are removed from the atmosphere mainly via chemical oxidation. Dry deposition is a minor sink, and wet deposition is almost negligible. The gas-phase chemistry of aromatics has been the subject of many studies (e.g., Atkinson et al., 1989; Warneck, 1999; Koppmann, 2007). Aromatics have relatively atmospheric lifetimes ranging from a few hours, e.g. for trimethylbenzene, to about ten days, e.g. for benzene (Atkinson and

Arey, 2003). Their oxidation is mainly controlled by the OH radical but they also react with $NO_3$ and $O_3$. The reaction with OH can proceed along two principal pathways. The first starts with H-abstraction from an aliphatic substituent. The following reactions are similar to those of aliphatic compounds and involve the addition of $O_2$, yielding a peroxy radical as an intermediate. Toluene, for example, can be oxidized in this way to benzaldehyde (Atkinson and Arey, 2003, and references therein). The second, which is the dominant path, is OH addition to the aromatic ring. Secondary reactions can lead to ring opening

and complex further reactions, eventually generating HCHO, glyoxal, and other smaller organic molecules (Vereecken, 2019, and references therein). The products from the oxidation of aromatic compounds have a reduced volatility and allow for the formation of SOA (Henze et al., 2008; Lin et al., 2012), which in turn can significantly reduce the gas-phase concentrations of the aromatic oxidation products.

Numerical models are essential to understand the highly complex chemical degradation of aromatics and to quantify the

impact of these compounds in atmospheric chemistry. A very detailed modeling of aromatics is possible with the reactions contained in the Master Chemical Mechanism (MCM, Jenkin et al., 2003). However, due to its complexity, the full mechanism

is mainly suitable for box model calculations. For global studies, simplified reaction schemes are usually used (e.g., Emmons et al., 2010; Hu et al., 2015).

The main objective of this study is to investigate how tropospheric OH, $O_3$, $NO_x$, and several VOC concentrations are affected by the oxidation of several monocyclic aromatics. The paper is organized as follows. In Sect. 2, the numerical model and the set-up of the simulations are described. Section 3 analyzes the calculated impact on selected chemical species both on the global and on the regional scales.

## 2 Model description

We used the ECHAM5/MESSy Atmospheric Chemistry (EMAC) model, which is a numerical chemistry and climate simulation system that includes submodels describing tropospheric and middle atmosphere processes (Jöckel et al., 2010). EMAC uses the second version of the Modular Earth Submodel System (MESSy2) to link multi-institutional computer codes. The core atmospheric model is the 5th generation European Centre Hamburg general circulation model (ECHAM5, Roeckner et al., 2006).

For the present study we performed simulations with EMAC (ECHAM5 version 5.3.02, MESSy version 2.53) in the T63L31ECMWF resolution, which corresponds to a grid with a horizontal cell size of approximately $1.875° \times 1.875°$ and 31 vertical hybrid pressure levels, extending from the surface up to about 10 hPa.

Emission rates of the individual aromatics are shown in Table 1. The sum of all sources is 29.4 TgC/a. For anthropogenic emissions, we used EDGAR 4.3.2 (Huang et al., 2017), distributed vertically as in Pozzer et al. (2009). The MESSy submodel MEGAN calculates biogenic emissions (Guenther et al., 2012). For biomass burning, the submodel BIOBURN was used, which integrates the Global Fire Assimilation System (GFAS) inventory (Kaiser et al., 2012).

Atmospheric chemistry was calculated with the MECCA submodel, which has been evaluated by Pozzer et al. (2007) and Pozzer et al. (2010). The most recent model version has been described by Sander et al. (2019). The mechanism for aromatic species is a reduced version of the MCM (Bloss et al., 2005b), as described in detail by Cabrera-Perez et al. (2016). In short, the MCM schemes for benzene and toluene were taken. Following the approach of Taraborrelli et al. (2009), short-lived intermediates were replaced with their stable products and isomeric peroxy radicals were lumped preserving the yield of stable products. Initial oxidation steps of aromatics other than benzene and toluene are considered and products replaced by the analogous toluene oxidation products. This approximation inherently introduces an error with respect to the formation of larger and low volatile products. The carbon mass that is not accounted for with this approximation is however tracked by introducing the counter LCARBON for the difference of carbon atoms between the oxidation products of larger aromatics and toluene. In this study, we consider several additions to the MCM reactions:

- For several nitrophenols (MCM names: HOC6H4NO2, DNPHEN, TOL1OHNO2, MNCATECH, DNCRES), their photolytic production of HONO were added (Bejan et al., 2006), e.g.:

$$+ \; HONO + CO_2 + CO + 2 \; HO_2 \quad \text{(R1)}$$

In JVAL (Sander et al., 2014) the cross sections for 2-nitrophenol and 3-methyl-2-nitrophenol and the quantum yield for 2-nitrophenol by Chen et al. (2011) are used to calculate the $j$-values.

- For the photolysis of benzaldehyde, the MCM uses the rate constant ($j$-value) of methacrolein as a proxy. Our model calculates with JVAL (Sander et al., 2014) the $j$-value based on the UV/VIS spectrum of benzaldehyde recommended by Wallington et al. (2018). In our code, the photolysis of benzaldehyde produces $C_6H_5O_2$, $HO_2$ and $CO$.

- For several phenyl peroxy compounds (MCM names: C6H5O2, CATEC1O2, OXYL1O2, MCATEC1O2, NCRES1O2), their reactions with $NO_2$ were added (Jagiella and Zabel, 2007), e.g.:

(R2)

- For the reaction of $HO_2$ with the peroxy radical C6H5CO3 (resulting from the oxidation of benzaldehyde), we use the yields provided by Roth et al. (2010).

- Alkyl nitrate yields are calculated as a function of temperature and pressure, as described by Sander et al. (2019).

- Bicyclic peroxy radicals in the oxidation mechanism of toluene yield 60% glyoxal and 40% methyl glyoxal from the non-radical terminating reactions with NO and $HO_2$ as suggested by Birdsall et al. (2010). Benzene is treated analogously but yields 100% glyoxal from the above mentioned reactions.

The aerosol calculations follow the approach of Pringle et al. (2010), with the notable difference of the inclusion of the explicit organic aerosol submodel ORACLEv1.0 by Tsimpidi et al. (2014). Although, similar to Tsimpidi et al. (2014), low- and intermediate volatiles are parameterized as lumped species, the equilibrium with their equivalent aerosol phase is explicitly

calculated for $\simeq 600$ volatile organic carbon tracers via ORACLE. The volatility and the enthalpy of vaporization of each tracer is estimated with the approaches of Li et al. (2016) and Epstein et al. (2010), respectively.

The simulated period covers the years 2009–2010, with the first year as spin-up, and the year 2010 being used for the analysis. The feedback between radiation and chemistry was decoupled to avoid any influence of chemistry on the dynamics (QCTM mode by Deckert et al. (2011)). As a consequence, every simulation discussed here has the same meteorology, i.e., binary identical transport.

To analyze the influence of the aromatic compounds on atmospheric chemistry and composition, we performed three model simulations, as listed in Table 2. The *AROM* simulation includes all chemical reactions and emissions of the following mono-cyclic aromatic compounds: benzene, toluene, xylenes (lumped), phenol, styrene, ethylbenzene, trimethylbenzenes (lumped), benzaldehydes, and higher aromatics (as representative of aromatics with more than 9 carbon atoms). The reference simulation (*NOAROM*) is identical to *AROM*, except that it excludes aromatic compounds. In the *ONLYMCM* run, we reverted the additions and changes to the MCM that have been described above. Our focus is to compare *AROM* with *NOAROM*. Results of *ONLYMCM* are mainly interesting for benzaldehyde and HONO. As EMAC uses terrain-following vertical hybrid pressure coordinates, we will refer to "surface" as the lowest model level, with an average thickness of roughly 60 m.

## 3    Results and discussion

Globally averaged surface mixing ratios obtained from all model simulations (*AROM*, *NOAROM*, and *ONLYMCM*) are listed in Table 3. Figure 1 shows the annual average mixing ratios of the sum of all aromatic compounds included in the simulation *AROM*. They are higher in continental areas and close to the surface. The highest values are predicted in the northern hemisphere (NH), in particular, in East and South Asia, as well as in parts of Europe, Africa, and the US, reaching up to about 1 nmol/mol. The background mean mixing ratios in oceanic areas of the southern hemisphere (SH) are of the order of a few pmol/mol. For a more detailed analysis, we have selected the following five regions, as defined in Figure 2: Amazon area (AMA), central Africa (CAF), eastern Asia (EAS), Europe (EUR), and eastern US (EUS). The budgets of selected chemical species were calculated within these regions (Table 5).

### 3.1    Hydroxyl radical (OH)

Figure 3 shows the model-calculated surface OH in the *AROM* and *NOAROM* simulations. When aromatics are introduced to the model, the global average concentration of OH decreases for two reasons: first, the direct reaction with aromatics consumes OH, and second, additional CO resulting from the degradation of aromatics represents an increased sink for OH. However, in eastern Asia, Europe, and the east coast of the US, where $NO_x$ concentrations are high, an increase of OH can be seen. Although the aromatics decrease $NO_x$ in these areas (see below), the chemical system remains in the high-$NO_x$ regime.

We find that inclusion of aromatics emissions leads to an increase OH in these regions but to decrease in the low-$NO_x$ CAF region. The increased OH in the high-$NO_x$ regions is mainly caused by the reaction of NO with $HO_2$. The production of OH from this important reaction is enhanced by the significant $HO_2$ formation in aromatics oxidation. Compared to *onlyMCM* the

*AROM* simulation has additional $HO_2$ production from the photolysis of ortho-nitrophenols (R1) and benzaldehyde (Sect. 2). The enhanced $HO_2$ levels (not shown) overcompensates the negative changes in NO (see Sect. 3.3).

Figure 4 shows the seasonal cycle of the OH mixing ratio in the planetary boundary layer for the NH and SH. Inclusion of the aromatics leads to a relative decrease between 2.5 % and 5.5 %. Higher OH concentrations are identified over continental areas during the NH autumn, winter and spring than in summer (Fig. 3). In summer, OH concentrations increase only at a few locations when aromatics are included. In general enhancements are predicted for regions where radical production is not $NO_x$-limited. In the NH there obviously more such regions compared to the SH. However, the largest decrease in the planetary boundary OH is computed for the NH where most of the emissions of aromatics are located.

Figure 5 shows the annual zonal mean changes of the OH mixing ratio. The changes are most pronounced in the NH upper troposphere where reductions range from 7 % to 20 %. These predicted changes are associated to similar reductions in $NO_x$. In fact, the upper troposphere is in general $NO_x$-limited and the oxidation of aromatics enhances the formation $N_2O_5$ and $HNO_3$ which are lost heterogeneously. This leads to an effective removal of $NO_x$ from the gas phase and lowers the radical production. The change in hemispheric burdens of OH are consistent with this picture (Table 4). This moderately helps bringing the model-simulated inter-hemispheric OH asymmetry closer to that derived from observations (Lelieveld et al., 2016). Globally, aromatics oxidation reduces OH by 7.7 % and consequently increases methane lifetime by about 5.5 %. The changes are more pronounced in the northern hemisphere where aromatics are mostly emitted (Table 4). However, the latter in the EMAC model remains significantly lower than the ACCMIP multi-model mean and the observational-based estimates (Naik et al., 2013). Coarse model spatial resolutions (about 200 km) are known to result in an overestimation (underestimation) of global mean OH (methane lifetime) of at least 5 % (Yan et al., 2016). This is due to a less efficient conversion of $NO_x$ to $NO_y$ when strong pollutant emissions are artificially diluted in the model grid boxes. This aspect certainly has a larger impact on the inter-hemispheric OH asymmetry in atmospheric models that is in contrast to observational estimates (Patra et al., 2014).

Differences for OH between the *AROM* and *onlyMCM* simulations are shown in Figures A1 and A2 of the Appendix A.

## 3.2 Ozone ($O_3$)

In most areas of the globe, surface ozone is slightly lower in *AROM* than in *NOAROM* (Fig. 6). The $O_3$ reduction is due to (i) the decrease in $NO_x$ concentrations (limiting ozone formation) and (ii) increasing radical production ($HO_x$, and $RO_2$) in ozone-depleting regimes, which enhances reaction of $O_3$ with $HO_2$. Only a few high-$NO_x$ regions, where hydrocarbons are the limiting factor for ozone formation, show increased ozone concentrations: mainly East China (EAS), but also the eastern US (EUS) and Europe (EUR). The increases in these areas is associated with anthropogenic emissions of aromatics, which have significant ozone formation potentials. We find that anthropogenic emissions of aromatics leads to an increase of $O_3$ in the EAS and EUR regions but to a decrease in the low-$NO_x$ CAF region.

The seasonal cycles of the relative differences show lower amplitude than for OH, but similar patterns (Fig. 7). The impact of aromatics is smallest in summer. Like for the OH levels, the inter-hemispheric asymmetry in the emission of aromatics determines the higher$O_3$ decrease in the NH compared to the SH.

The zonal mean changes of $O_3$ mixing ratio in the troposphere are uniformly negative (Fig. 8). Similar to surface ozone, the annual mean changes for *ONLYMCM* and *AROM* are $-2.3\%$ and $-3.0\%$, respectively. The hemispheric changes are shown in Table 4. It is well known that MCM for aromatics overestimates ozone production in chamber experiments (Bloss et al., 2005b). The issue has been analysed in the companion paper (Bloss et al., 2005a) where the best mechanism improvement was found to be an early OH source during oxidation. Cabrera-Perez et al. (2016) introduced enhanced $HO_x$-sources by photolysis of benzaldehyde and nitrophenols. These modifications consistently result in less ozone produced with respect to MCM. These results deviate from the results by Yan et al. (2019) who suggested a global increase of 0.4 % due to aromatics. However, they only considered benzene, toluene and xylenes. Our results, obtained with a more comprehensive setup, suggest that aromatics could slightly ameliorate the model overestimate in the NH (Jöckel et al., 2016; Young et al., 2018). The overall tropospheric ozone burden decreases from 381 to 369 Tg for the *AROM* simulation. These estimated changes are robust against the tropopause definition and are about -3.5 and -2.3 % for the Northern and Southern Hemispheres, respectively (Table 4). The changes in ozone are caused by perturbations of the radical production in different $NO_x$ regimes but also by the direct ozone loss in reactions with organic compounds. It is widely acknowledged that this direct loss is only due to the ozonolysis of unsaturated VOCs and is estimated to be about 100 $Tg/yr$, less than 2 % of the tropospheric ozone budget (e.g. in Tilmes et al. (2016)). However, with aromatics a new direct ozone loss process involving organic radicals comes in place. In Figure 9 the change in tropospheric ozone burden is shown against the change in ozone loss with organic compounds. This change is estimated to be globally in the 200-300 Tg/yr range depending on the mechanism used and is comparable to the loss by bromine chemistry in the troposphere (Sherwen et al., 2016). Ozone is known to react with organic radicals like methyl peroxy radical although this loss is an insignificant sink (Tyndall et al., 1998). We find that (substituted) phenoxy radicals from aromatics are a significant sink term of ozone ($>200$ $Tg/yr$). These radicals are unique to aromatics oxidation and they also react with NO and $NO_2$. When the concentrations of $NO_x$ are relatively low, $C_6H_5O$ has sufficiently long lifetime to react with $O_3$. This ozone loss is modelled based on the results by Tao and Li (1999) for phenoxy radical:

(R3)

Although the known rate constant for reaction R3 is about one order of magnitude lower than the others, the high abundance in the atmosphere makes ozone the major sink of (substituted) phenoxy radicals. This direct ozone loss in reaction R3 is enhanced by phenoxy radical production in reaction R2 and the concurrent loss of odd oxygen by $NO_3$ photolysis and $N_2O_5$ heterogeneous loss

$$NO_3 + h\nu \quad \rightarrow \quad NO + O_2 \tag{R4}$$

$$NO_3 + NO_2 \quad \rightarrow \quad N_2O_5 \tag{R5}$$

$$N_2O_5 + H_2O \quad \rightarrow \quad 2HNO_3(aq) \tag{R6}$$

In our chemical kinetics mechanism (also in MCM) the reaction system just described constitutes an effective catalytic destruction cycle of odd oxygen. The strength of this cycle has not been diagnosed in this study. Nevertheless, we observe that it depends on the (substituted) phenoxy radical levels and is significantly reduced in *AROM* compared to *onlyMCM* (Figure 9). We ascribe this difference to MCM not accounting for the photolysis of nitrophenols (R1) as determined by Bejan et al. (2006). In fact, in MCM the first nitrophenols from benzene (HOC6H4NO2) and toluene (TOL1OHNO2) solely form nitrophenoxy radicals with the same reactivity of the unsubstituted phenoxy radical (C6H5O). Thus, the photolysis of nitrophenols decreases the amount of ozone lost by reaction with nitrophenoxy radicals. The impact of all the additions and modifications to the MCM on the predicted $O_3$ levels is shown in Figures A3, A4 of the Appendix A. Uncertainties on the reactions mentioned in this paragraph are discussed in Section 4.

Our results for ozone differ both in magnitude and sign compared to the global study by Yan et al. (2019). However, the latter used the SAPRC-11 oxidation mechanism (Carter and Heo, 2013) which does not account for the reaction of phenoxy radicals with ozone (R3) and phenylperoxy radicals with $NO_2$ (R2).

### 3.3 Inorganic nitrogen

The simulated annual mean $NO_x$ concentrations at the surface are significantly lower in *AROM* than in *NOAROM* (Figs. 10 and 11). One reason is the formation of aromatic species containing nitrogen (e.g., nitrophenols) in *AROM*, thereby transferring part of the $NO_x$ burden to the nitrogenated species. The largest decreases (both absolute and relative) are found in regions with high $NO_x$ concentrations. Since the ozone chemistry is not $NO_x$-limited in these regions, the impact on ozone is small. This holds for the free troposphere for which zonal average decreases in $NO_x$ can be larger than 20 % (not shown), which in turn significantly influence OH (Fig. 5).

On the one hand, the reaction with aromatics is a sink for $NO_3$. On the other hand, $NO_3$ is produced in the phenylperoxy reaction with $NO_2$ (R2). However the latter seems to dominate and cause a significant and widespread increase in the predicted $NO_3$ levels. Relative to *NOAROM*, in *AROM* the global average of the nighttime species $NO_3$ increases by more than 7 % (Table 3). In contrast to the global mean tendency, $NO_3$ modest decreases in several regions in Africa, South America, and India (Fig. 12). These decreases correlate well with emissions from biomass burning. Differences for $NO_3$ between the *AROM* and *onlyMCM* simulations are shown in Figures A5 of the Appendix A.

Although the net change of global HONO is small (about 3 % less in *AROM* than in *NOAROM*, see Figure 13 and Table 3), the regional differences can be large (Table 5). A decrease of HONO is seen mainly in polluted areas (EAS, EUR, EUS) in the winter. In contrast, HONO increases in the regions with emissions from biomass burning (AMA, CAF). Here, HONO is

formed by the photolysis of nitrophenols (R1). Since these reactions are not included in the MCM, we do not see any HONO increase in the *ONLYMCM* simulation (Fig. 14).

On a global average level, $HNO_3$ is not affected much by aromatics. However, an increase can be seen in the regions where ozone increases (EAS) or where biomass burning decreases $NO_3$ and $N_2O_5$ (CAF), see Figure 15 and Table 5. An average zonal mean change of up to 5% throughout the UT/LS is linked to the enhanced $NO_3$ production by R2.

### 3.4 Selected oxygenated compounds

Globally, HCHO is not affected much by aromatics. There are, however, regional differences that are moderate because of the concurrent enhancement of the HCHO sink by reaction with OH. We find maximum absolute depletions in the AMA region, where concentrations are typically high (Fig. 16). Increased values of HCHO are mainly seen in EAS and EUR (Table 5).

$\alpha$-dicarbonyls like glyoxal and methyl glyoxal are primarily produced from the bicycloalkyl-radical pathway leading in the case of benzene to BZBIPERO2 (MCM) (Volkamer et al., 2001). A minor secondary formation pathway from conjugated unsaturated dicarbonyls, e.g., MALDIAL (MCM), is also known and taken into account (Bloss et al., 2005b). As expected, the model predicts a very large increase of glyoxal in almost all continental areas (Figs. 17 and 18). The global burden is 36 % higher than in the *NOAROM* model simulation. The largest regional increases are in the EAS and EUR regions (Table 5). An exception to the global trend is the AMA region, where OH is too low to produce either glyoxal or methyl glyoxal. Annual mean increases exceed 50 % over the continents close to the surface. In the lower troposphere, zonal mean increases are in the 10-20 % range. These changes are of significance for the model SOA budget since these two dicarbonyls are estimated to produce a large fraction of SOA by cloud processing yielding low-volatile oligomers (Lin et al., 2012). However, a model assessment of SOA formation from $\alpha$-dicarbonyls is beyond the scope of this study. The reason is that, although the simulations were performed with a VBS-based approach to model condensation of organic vapours, the EMAC model version used in this study has no representation of oligomer formation from (methyl)glyoxal. This has been recently implemented explicitly for cloud droplets (Rosanka et al., 2020) and its effect is planned to be assessed in a subsequent study together with the contribution of reactive uptake of epoxides from isoprene and aromatics.

Comparing *AROM* to *ONLYMCM*, benzaldehyde decreases by more than 50 % when the photolysis rate constant ($j$-value) from the MCM (based on methacrolein) is replaced by our value (based on the UV/VIS spectrum of benzaldehyde). The more realistic photolysis rate enhances the production of radicals like $HO_2$.

Since additional reactive carbon compounds have been introduced in the model, the oxidation of aromatics produces more CO, which has a lifetime of about 1-3 months (Lelieveld et al., 2016). CO can travel long distances from its source, although its lifetime is not long enough to allow it to cross hemispheres (Daniel and Solomon, 1998). CO concentrations generally increase on the global scale, indicating a small addition to the carbon budget. When comparing *AROM* to *NOAROM*, we find an increase of about 6 % in the atmospheric burden of CO. Interestingly, maximum zonal average increases of 10-20 % are found for the NH upper troposphere/lower stratosphere (UTLS) region (Fig. 19).

## 4 Model uncertainties

The model calculations presented in this work are associated with some uncertainties related to the oxidation kinetic model, emissions and model resolution.

Gas-phase oxidation of aromatics is complex and the kinetic mechanism used in this study reflects the state of knowledge, advancements and limitations in the mechanism have recently been discussed (Vereecken, 2019). Recent progress has focused in particular on the source strength of aerosol precursors and not on the overall radical production which also affects ozone. Nevertheless, our kinetic model makes use of only one rate constant for the reaction R3 of phenoxy radicals with ozone (Tao and Li, 1999). It also assigns this rate constant to the substituted phenoxy radicals other than $C_6H_5O$. Unfortunately, there is only one study of the rate constant of R3 at 298 K. Although the 2-$\sigma$ reported uncertainty is slightly larger than 10 %, the rate constant of $2.86 \times 10^{-13}$ $cm^3$ $molecules^{-1}$ $s^{-1}$ has to be regarded as a lower limit. On the other hand, experimental evidence for the product of R3, being phenyl peroxy radical ($C_6H_5O_2$), has not been found although it was expected. If the products are different, then the catalytic $O_3$-destruction cycle illustrated in Sec. 3.2 would not be in place. However, a significant amount of ozone loss via R3 and analogous reactions is to be expected. Moreover, the ozone loss is likely underestimated because of the model not accounting for the photolysis of nitrophenols forming nitrosophenoxy radicals. Different from the HONO-formation channel, which destroys the aromatic ring, channels yielding substituted phenoxy radicals may dominate (Cheng et al., 2009; Vereecken et al., 2016) and thus enhance ozone loss. Another source of uncertainty is the direct formation of epoxide upon addition of OH and subsequently by $O_2$ as implemented in the MCM ranging from 11.8 %, for benzene, to 24 %, for trimethylbenzene (Bloss et al., 2005b,a). There in fact consistent theoretical evidence that the epoxide formation pathway passes through a second $O_2$-addition. This implies that the epoxide yield likely depends on the abundance of $NO, HO_2 and RO_2$ (Vereecken, 2019, and references therein). This uncertainty limit the reliability of the predicted SOA formation from reactive uptake of epoxides by aerosols (Paulot et al., 2009).

Cloud chemistry of organic compounds is known to suppress gas-phase $HO_x$-production and directly consume ozone (Lelieveld and Crutzen, 1990). The overall effect on ozone depends on the local chemical regime. In our study water-soluble products are set to only undergo wet deposition (dissolution and removal by precipitation). Their aqueous-phase chemistry might however have a non-negligible effect on ozone and other oxidants. For instance, phenol is known to react very quickly with OH in the aqueous-phase (Field et al., 1982). Moreover, phenoxide anions from phenols react quickly with ozone (Hoigné and Bader, 1983). In particular, nitrophenols might be efficient ozone scavengers as they are stronger acids than unsubstituted phenols. A global assessment of cloud chemistry involving aromatics oxidation products is possible with the modelling system used here (Tost et al., 2006, 2010). However, considering the complexity of aqueous-phase oxidation of organic compounds, such an assessment is outside the scope of this study and deserves a dedicated model study.

In our study, biomass burning emissions of aromatics are potentially underestimated. In fact, based on the recent update by Andreae (2019), we estimate that emissions might be up to 5 $Tg/a$ (65%) higher than what is implemented in our model. Moreover, emissions from peat fires in 2010 (the simulated year) were up to a factor 15 lower than in the subsequent years (van der Werf et al., 2017). In general, the inter-annual variability of biomass burning is large and difficult to capture in a study

such as the present one. However, it appears that the two major contributions to this variability are the peat fires in Indonesia and boreal forest fires, which are strongly favoured by El Nino and heat waves, respectively. An early estimate of anthropogenic emissions of aromatics gave 16 TgC/a, (Fu et al., 2008). Two relatively recent datasets yield about 50% higher emissions being 23 TgC/a for RCP (Cabrera-Perez et al., 2016) and 22 TgC/a for EDGAR 4.3.2 (Huang et al., 2017). The latter is used in this study and lacks the biofuel burning emissions of phenol, benzaldehyde and styrene. Inter-annual variability of anthropogenic emissions of aromatics is is not well known but the decadal trends are known to be negative since the 1980s (Lamarque et al., 2010). Aromatics emissions from terrestrial vegetation have been long neglected or considered very low. However, Misztal et al. (2015) suggested that aromatics emissions from biogenic sources may rival those from anthropogenic ones. In this study we used the same emission algorithm used in Misztal et al. (2015) but get much lower emissions for toluene (about 0.3 vs. 1.5 TgC/a). However, Misztal et al. (2015) suggest that emissions of aromatics and benzenoid compounds may be in the 1.4-15 TgC/a range. The major contributors are toluene and some benzenoids (oxygenated aromatics). The latter are mainly emitted during blossoming and stress-induced reactions by plants. The variability of their emissions is not very well quantified. For instance, the MEGAN model calculates their emission strengths based of the ones for carbon monoxide (Tarr et al., 1995).

The spatial resolution of atmospheric models has a significant influence on the predicted levels of oxidants and nitrogen oxides. Generally in polluted regions the coarser the resolution the larger the ozone production per molecule of $NO_x$ will be (Sillman et al., 1990). This is due to the artificial dilution of strong NOx emissions which, in reality, is efficiently converted to $NO_y$ by reacting with $HO_x$. For instance, reducing the spatial resolution over the polluted North America, Europe and East Asia with a two-way nested regional model led to a 9.5 % reduction in the global tropospheric ozone burden (Yan et al., 2016). We have shown that at our model resolution of $1.875° \times 1.875°$ aromatics are estimated to induce important increases in $HO_x$ (Fig. 3) and decreases in $NO_x$ (Fig. 10 and 11) over continental polluted regions. Therefore, at much higher spatial resolutions we expect that the enhancement of surface ozone by aromatics in those regions (Fig. 6) to be greatly reduced if not reverted. Based on the results by Yan et al. (2016) we expect this effect to translate in a significant enhancement of the tropospheric ozone reduction reported in this study (Sect. 3.2). A quantification of the model resolution effect on chemical regimes is at the moment computationally prohibitive with our very large chemical scheme running in the global EMAC model.

Finally, atmospheric levels of benzene and toluene simulated by our model were shown to underestimate many observations by at least 20% (Cabrera-Perez et al., 2016). It is worth noting that in Cabrera-Perez et al. (2016) the total emissions of aromatics were even slightly higher (2.6 $\mathrm{TgC/yr}$) than in the *AROM* simulation. This underestimate could be explained by an overestimate of the chemical sink in the troposphere by reaction with hydroxyl radical. However, the annual global mean concentration of hydroxyl radicals is potentially 10% too high (Lelieveld et al., 2016), which cannot account for model concentration biases that are larger than 20%. Therefore, we surmise that the impact of aromatics on the trace gas composition may be larger than estimated in this study.

## 5 Summary

This study investigates the effects of several monocyclic aromatics on the tropospheric gas-phase composition by means of the chemistry-climate model EMAC. When aromatics are introduced into our model calculations, large changes are seen for glyoxal and methyl glyoxal. For other species, our results show a relatively small importance of aromatics on the global scale. This is consistent with recent results by Yan et al. (2019) who used a simpler chemistry mechanism in the GEOS-Chem model. However, different from that study, we found a negative impact on global ozone. Our results also indicate that by including aromatics chemistry, free tropospheric OH is reduced, especially in the northern hemisphere. On a regional scale, the concentrations of several species change significantly, with relatively largest impacts in East Asia where emissions are higher. Regions with high $NO_x$ concentrations show increases of OH and $O_3$. However, since these increases are counteracted by decreases downwind, i.e., in remote areas where $NO_x$ concentrations are much lower, the net effects on large scales are small. Of the nitrogen compounds, mainly $NO_3$ and HONO are affected by the aromatics chemistry.

We conclude that, although the impact of aromatics is relatively minor on the global scale, it is important on regional scales, notably in the anthropogenic source regions, and especially in those where $NO_x$ emissions are strongest. Given the uncertainties in the oxidation mechanisms and emissions, the results of our model calculations may underestimate the impact of aromatics on the tropospheric gas-phase composition.

*Code availability.* The Modular Earth Submodel System (MESSy) is continuously further developed and applied by a consortium of institutions. The usage of MESSy and access to the source code is licensed to all affiliates of institutions which are members of the MESSy Consortium. Institutions can be a member of the MESSy Consortium by signing the MESSy Memorandum of Understanding. More information can be found on the MESSy Consortium web-page (http://www.messy-interface.org).

*Author contributions.* DT, RS, AP and DC wrote the manuscript. AP and DC performed the model simulations. DT and RS developed and analyzed the chemical mechanism. SB visualized the model results. SG performed extended budgeting of species' chemical turnover. All co-authors contributed to the analysis of results and the writing of the paper.

*Competing interests.* The authors have no competing interests

*Acknowledgements.* The authors want to acknowledge the use of the Ferret program for analysis and graphics in this paper. Ferret is a product of NOAA's Pacific Marine Environmental Laboratory (information is available at http://www.ferret.noaa.gov). The work described in this paper has received funding from the Initiative and Networking Fund of the Helmholtz Association through the project "Advanced Earth System Modelling Capacity (ESM)" (information is available at https://www.esm-project.net/).

**Table 1.** Global annual emission rates of aromatic compounds included in the model simulations and their relative contributions.

| Species | total (TgC/a) | anthro- pogenic (EDGAR) | biomass burning (BIOBURN) | biogenic (MEGAN) |
|---|---|---|---|---|
| Benzene | 4.417 | 70 % | 30 % | |
| Toluene | 5.888 | 82 % | 13 % | 5 % |
| Xylenes | 5.664 | 96 % | 4 % | |
| Ethylbenzene | 1.961 | 74 % | 26 % | |
| Benzaldehyde | 1.382 | 92 % | 6 % | 2 % |
| Phenol | 2.559 | 43 % | 57 % | |
| Styrene | 1.596 | 91 % | 9 % | |
| Trimethylbenzenes | 0.906 | 94 % | 6 % | |
| Higher aromatics | 4.980 | 48 % | 52 % | |

**Table 2.** Sensitivity studies.

| Simulation | Description |
|---|---|
| *AROM* | Aromatics are fully included |
| *NOAROM (reference)* | No aromatics (emissions switched off) |
| *ONLYMCM* | Only MCM reactions |

**Table 3.** Globally averaged area-weighted mixing ratios at the surface (annual averages for 2010). "ABSDIFF" denotes the absolute difference, (e.g., AROM-NOAROM), and "RELDIFF" the relative difference, (e.g., AROM/NOAROM-1).

| | NOAROM | ONLYMCM | AROM | AROM vs ONLYMCM | | AROM vs NOAROM | |
| | | | | ABSDIFF | RELDIFF | ABSDIFF | RELDIFF |
| | mol/mol | mol/mol | mol/mol | mol/mol | % | mol/mol | % |
|---|---|---|---|---|---|---|---|
| OH | $4.630\times10^{-14}$ | $4.472\times10^{-14}$ | $4.487\times10^{-14}$ | $1.557\times10^{-16}$ | 0.3482 | $-1.425\times10^{-15}$ | -3.078 |
| $O_3$ | $3.269\times10^{-8}$ | $3.220\times10^{-8}$ | $3.190\times10^{-8}$ | $-2.964\times10^{-10}$ | -0.9204 | $-7.888\times10^{-10}$ | -2.413 |
| NO | $3.029\times10^{-11}$ | $2.793\times10^{-11}$ | $2.609\times10^{-11}$ | $-1.843\times10^{-12}$ | -6.599 | $-4.203\times10^{-12}$ | -13.87 |
| $NO_2$ | $3.389\times10^{-10}$ | $3.314\times10^{-10}$ | $3.191\times10^{-10}$ | $-1.228\times10^{-11}$ | -3.706 | $-1.977\times10^{-11}$ | -5.834 |
| $NO_3$ | $1.004\times10^{-12}$ | $9.462\times10^{-13}$ | $1.080\times10^{-12}$ | $1.339\times10^{-13}$ | 14.15 | $7.599\times10^{-14}$ | 7.568 |
| HONO | $7.393\times10^{-12}$ | $7.260\times10^{-12}$ | $7.315\times10^{-12}$ | $5.538\times10^{-14}$ | 0.7628 | $-7.754\times10^{-14}$ | -1.049 |
| $HNO_3$ | $1.420\times10^{-10}$ | $1.393\times10^{-10}$ | $1.426\times10^{-10}$ | $3.352\times10^{-12}$ | 2.407 | $6.607\times10^{-13}$ | 0.4653 |
| HCHO | $5.993\times10^{-10}$ | $5.992\times10^{-10}$ | $6.002\times10^{-10}$ | $9.484\times10^{-13}$ | 0.1583 | $8.414\times10^{-13}$ | 0.1404 |
| glyoxal | $1.040\times10^{-11}$ | $1.444\times10^{-11}$ | $1.505\times10^{-11}$ | $6.117\times10^{-13}$ | 4.237 | $4.646\times10^{-12}$ | 44.67 |
| methyl glyoxal | $3.847\times10^{-11}$ | $4.005\times10^{-11}$ | $4.015\times10^{-11}$ | $1.051\times10^{-13}$ | 0.2625 | $1.682\times10^{-12}$ | 4.372 |
| benzaldehyde | | $6.798\times10^{-12}$ | $4.479\times10^{-12}$ | $-2.319\times10^{-12}$ | -34.11 | $4.479\times10^{-12}$ | |
| CO | $97.6\times10^{-9}$ | $103.3\times10^{-9}$ | $103.3\times10^{-9}$ | $-6.5\times10^{-11}$ | -0.06278 | $5.7\times10^{-9}$ | 5.847 |

**Table 4.** Simulated tropospheric integrals of OH, $O_3$ and $NO_x$, and the lifetime $\tau$ of $CH_4$. Tropospheric burdens were reckoned using six different tropopause definitions (provided by the TROPOP submodel, see Jöckel et al. (2010) for details): 1,2 surfaces of $O_3$ mixing ratio of 125 and 150 nmol/mol, respectively, 3) WMO definition (WMO (1957)), 4) dynamic PV-based (3.5 PVU potential vorticity surface, sought within 50–800 hPa), 5) climatological (invariable zonal profile, i.e. $300\text{-}215\bullet(\cos(\text{latitude}))^2$ hPa) and 6) the combined definition (WMO tropopause within 30°N–30°S, otherwise dynamic PV-based tropopause). The latter definition is used by default in EMAC and in this work. Estimated changes to tropospheric $O_3$ burden are identical within 0.05 % between the available definitions.

| | $n(OH)$ | | $m(O_3)$ | | $n(NO_x)$ | | $\tau(CH_4)$ | |
| Simulation | NH | SH | NH | SH | NH | SH | NH | SH |
|---|---|---|---|---|---|---|---|---|
| NOAROM | 6799 kmol | 5765 kmol | 207 Tg | 173 Tg | 7.90 Gmol | 4.02 Gmol | 7.36 yrs | 9.61 yrs |
| ONLYMCM vs NOAROM | −9.9 % | −7.3 % | −2.5 % | −2.1 % | −3.7 % | −1.0 % | +7.1 % | +4.7 % |
| AROM vs NOAROM | −9 % | −6.3 % | −3.5 % | −2.3 % | −10.8 % | −4.5 % | +6.8 % | +4.5 % |

**Table 5.** Regionally averaged mixing ratios of selected species (annual averages for 2010).

| | *NOAROM* mol/mol | *AROM* mol/mol | ABSDIFF mol/mol | RELDIFF % |
|---|---|---|---|---|
| | | OH | | |
| AMA | $2.861 \times 10^{-14}$ | $2.785 \times 10^{-14}$ | $-7.689 \times 10^{-16}$ | -2.687 |
| CAF | $6.447 \times 10^{-14}$ | $6.086 \times 10^{-14}$ | $-3.616 \times 10^{-15}$ | -5.608 |
| EAS | $4.712 \times 10^{-14}$ | $5.527 \times 10^{-14}$ | $8.147 \times 10^{-15}$ | 17.29 |
| EUR | $3.591 \times 10^{-14}$ | $3.852 \times 10^{-14}$ | $2.615 \times 10^{-15}$ | 7.283 |
| EUS | $5.629 \times 10^{-14}$ | $5.784 \times 10^{-14}$ | $1.553 \times 10^{-15}$ | 2.759 |
| | | $O_3$ | | |
| AMA | $2.979 \times 10^{-8}$ | $2.909 \times 10^{-8}$ | $-6.973 \times 10^{-10}$ | -2.341 |
| CAF | $3.856 \times 10^{-8}$ | $3.712 \times 10^{-8}$ | $-1.440 \times 10^{-9}$ | -3.733 |
| EAS | $3.124 \times 10^{-8}$ | $3.505 \times 10^{-8}$ | $3.807 \times 10^{-9}$ | 12.19 |
| EUR | $3.045 \times 10^{-8}$ | $3.033 \times 10^{-8}$ | $-1.250 \times 10^{-10}$ | -0.4105 |
| EUS | $3.930 \times 10^{-8}$ | $3.904 \times 10^{-8}$ | $-2.604 \times 10^{-10}$ | -0.6626 |
| | | $NO_3$ | | |
| AMA | $3.570 \times 10^{-13}$ | $3.483 \times 10^{-13}$ | $-8.678 \times 10^{-15}$ | -2.431 |
| CAF | $2.105 \times 10^{-12}$ | $2.321 \times 10^{-12}$ | $2.163 \times 10^{-13}$ | 10.27 |
| EAS | $1.833 \times 10^{-12}$ | $1.949 \times 10^{-12}$ | $1.163 \times 10^{-13}$ | 6.346 |
| EUR | $1.280 \times 10^{-12}$ | $1.256 \times 10^{-12}$ | $-2.448 \times 10^{-14}$ | -1.913 |
| EUS | $2.536 \times 10^{-12}$ | $2.488 \times 10^{-12}$ | $-4.802 \times 10^{-14}$ | -1.894 |
| | | HONO | | |
| AMA | $5.335 \times 10^{-11}$ | $5.349 \times 10^{-11}$ | $1.370 \times 10^{-13}$ | 0.2567 |
| CAF | $8.110 \times 10^{-11}$ | $8.227 \times 10^{-11}$ | $1.174 \times 10^{-12}$ | 1.447 |
| EAS | $1.152 \times 10^{-10}$ | $1.038 \times 10^{-10}$ | $-1.146 \times 10^{-11}$ | -9.945 |
| EUR | $5.689 \times 10^{-11}$ | $5.604 \times 10^{-11}$ | $-8.429 \times 10^{-13}$ | -1.482 |
| EUS | $4.415 \times 10^{-11}$ | $4.230 \times 10^{-11}$ | $-1.854 \times 10^{-12}$ | -4.199 |
| | | $HNO_3$ | | |
| AMA | $1.515 \times 10^{-10}$ | $1.508 \times 10^{-10}$ | $-7.056 \times 10^{-13}$ | -0.4657 |
| CAF | $4.957 \times 10^{-10}$ | $5.162 \times 10^{-10}$ | $2.048 \times 10^{-11}$ | 4.131 |
| EAS | $1.035 \times 10^{-9}$ | $1.169 \times 10^{-9}$ | $1.335 \times 10^{-10}$ | 12.89 |
| EUR | $3.985 \times 10^{-10}$ | $4.003 \times 10^{-10}$ | $1.855 \times 10^{-12}$ | 0.4656 |
| EUS | $6.706 \times 10^{-10}$ | $6.721 \times 10^{-10}$ | $1.505 \times 10^{-12}$ | 0.2244 |
| | | HCHO | | |
| AMA | $5.217 \times 10^{-9}$ | $5.189 \times 10^{-9}$ | $-2.874 \times 10^{-11}$ | -0.5509 |
| CAF | $3.468 \times 10^{-9}$ | $3.478 \times 10^{-9}$ | $9.392 \times 10^{-12}$ | 0.2708 |
| EAS | $1.322 \times 10^{-9}$ | $1.557 \times 10^{-9}$ | $2.348 \times 10^{-10}$ | 17.76 |
| EUR | $7.356 \times 10^{-10}$ | $7.708 \times 10^{-10}$ | $3.517 \times 10^{-11}$ | 4.781 |
| EUS | $1.911 \times 10^{-9}$ | $1.942 \times 10^{-9}$ | $3.096 \times 10^{-11}$ | 1.620 |

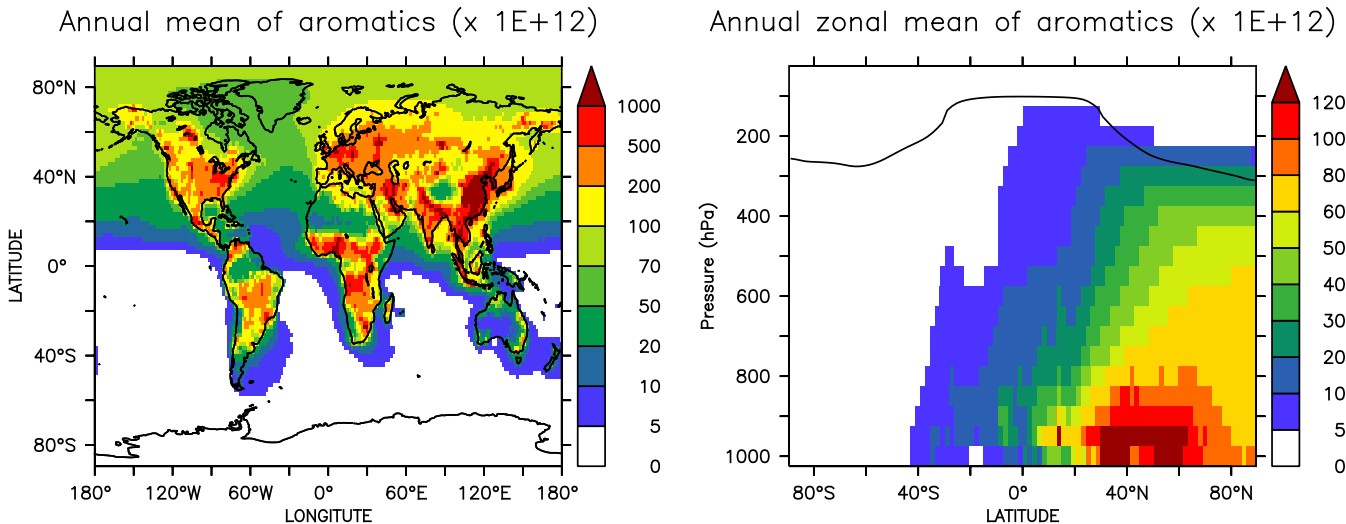

**Figure 1.** Annual mean mixing ratios of the sum of aromatics at the surface (left) and the zonal mean (right) in the *AROM* simulation. The solid line between 100 and 300 hPa depicts the mean tropopause level.

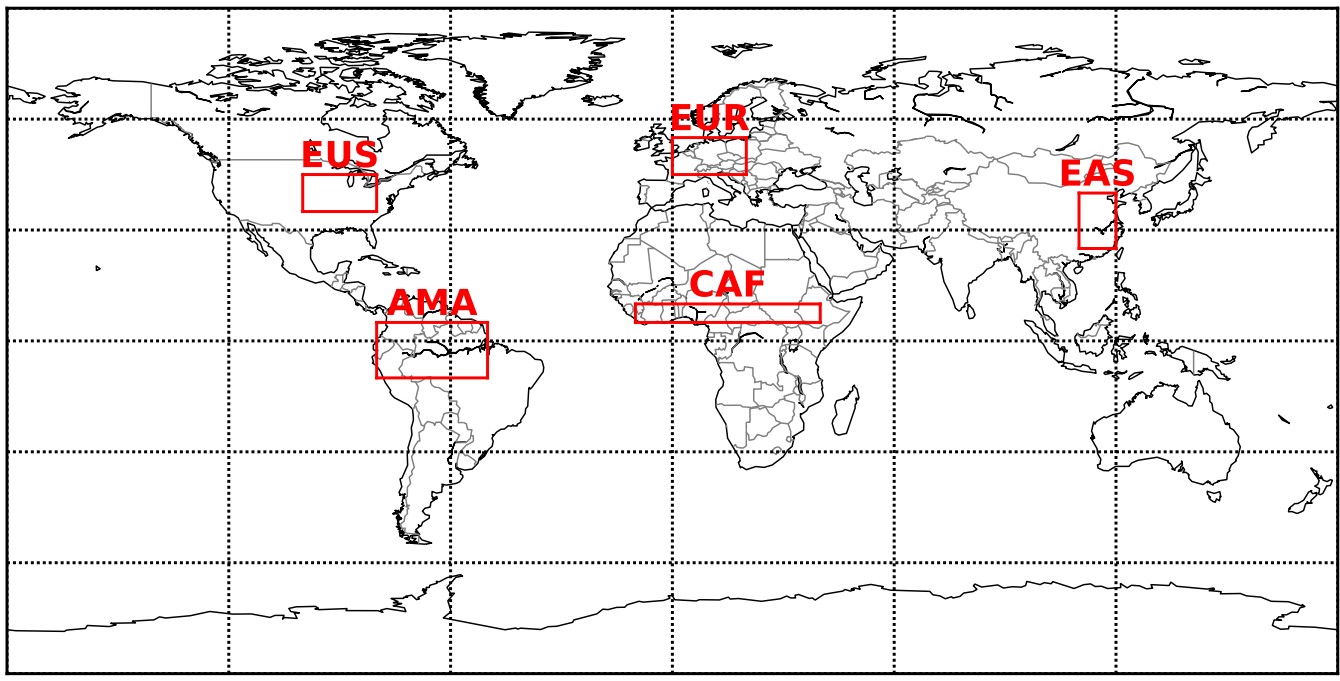

**Figure 2.** Selected regions: AMA = Amazon area, CAF = central Africa, EAS = eastern Asia, EUR = Europe, EUS = eastern US.

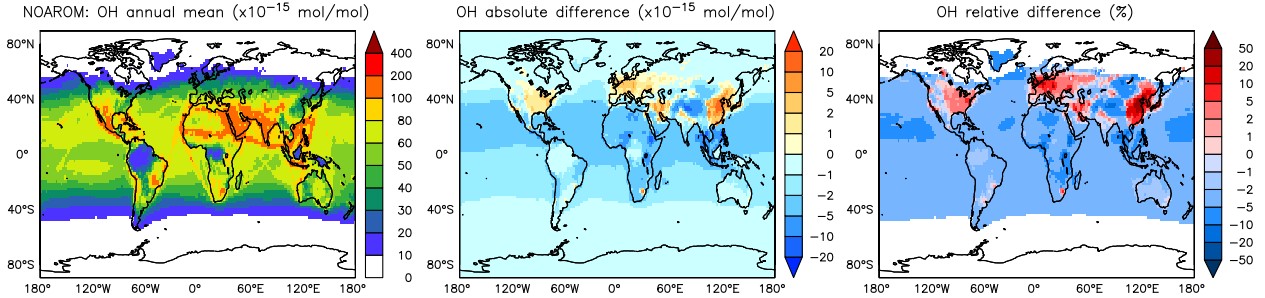

OH seasonal mean (x10⁻¹⁵ mol/mol)    OH absolute difference (x10⁻¹⁵ mol/mol)    OH relative difference (%)

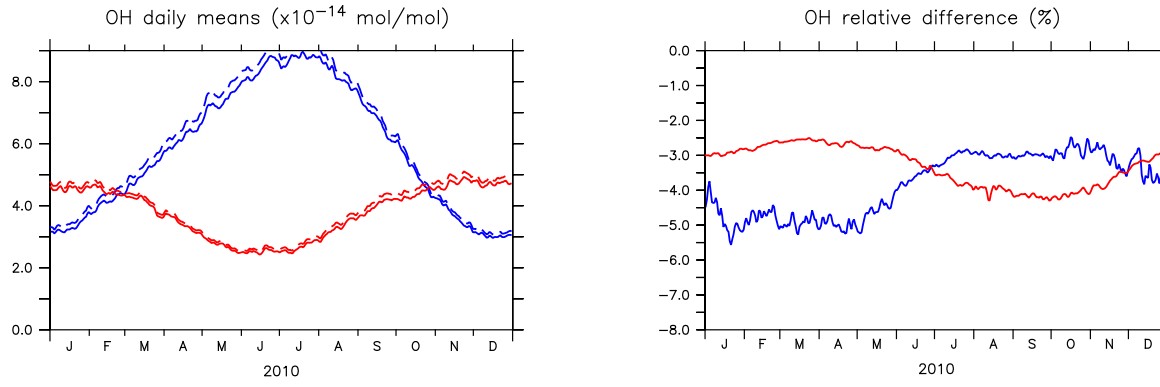

**Figure 4.** Left: Seasonal cycles of OH daily (24 h) mixing ratio means (in $10^{-14}$ mol/mol) in the planetary boundary layer (PBL) for AROM (solid line) and NOAROM (dashed line). Right: Relative difference (expressed in %) between AROM and NOAROM. In blue, values for the NH; in red, values for the SH. The PBL diagnosis is described in Pozzer et al. (2009). The PBL is calculated in the model based on the work of Holtslag et al. (1990). An interactive calculation is performed following the approach of Troen and Mahrt (1986), using the Richardson number, the horizontal velocity components, the buoyancy parameters and the virtual temperature (Holtslag and Boville, 1993).

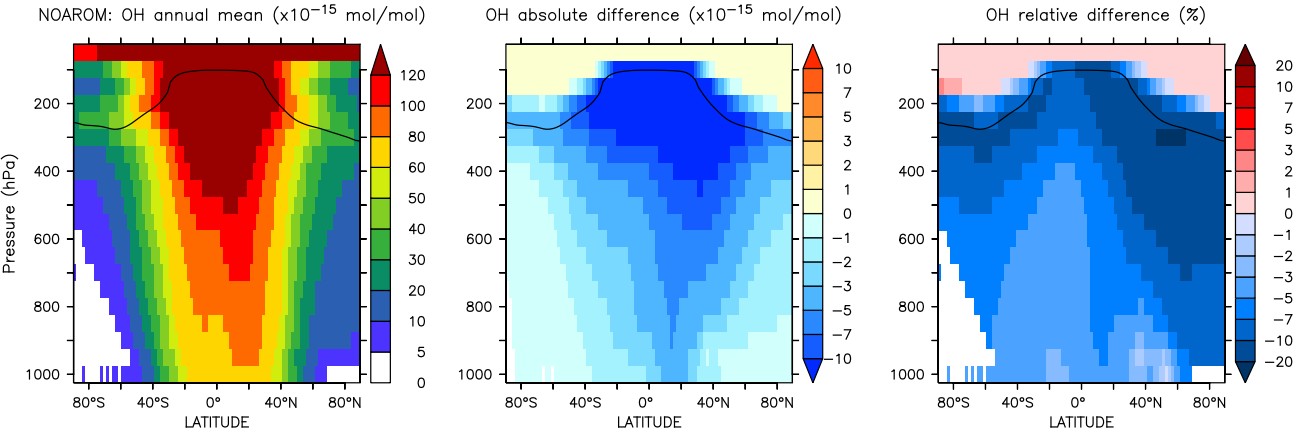

**Figure 5.** Annual average zonal mean OH mixing ratios. Left: Mixing ratios in the *NOAROM* simulation. Middle: Absolute difference *AROM-NOAROM*. Right: Relative difference *AROM/NOAROM*-1 in %. The solid line between 100 and 300 hPa depicts the mean tropopause level.

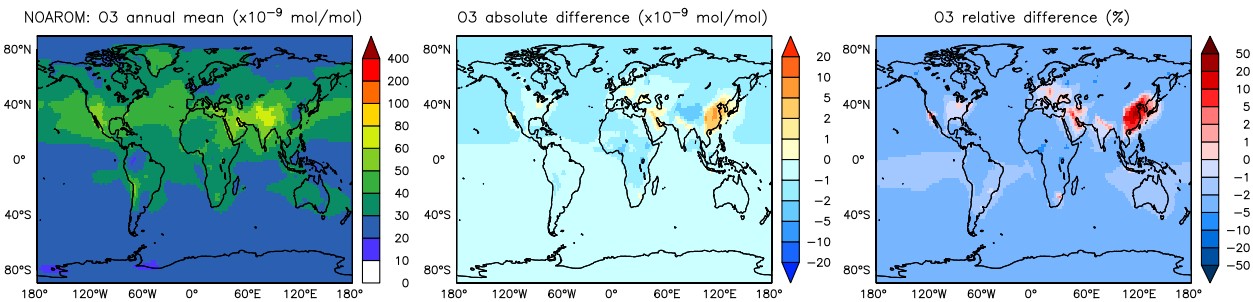

**Figure 6.** Annual average O₃ mixing ratios at the surface. Left: Mixing ratios in the *NOAROM* simulation. Middle: Absolute difference *AROM-NOAROM*. Right: Relative difference *AROM/NOAROM*-1 in %.

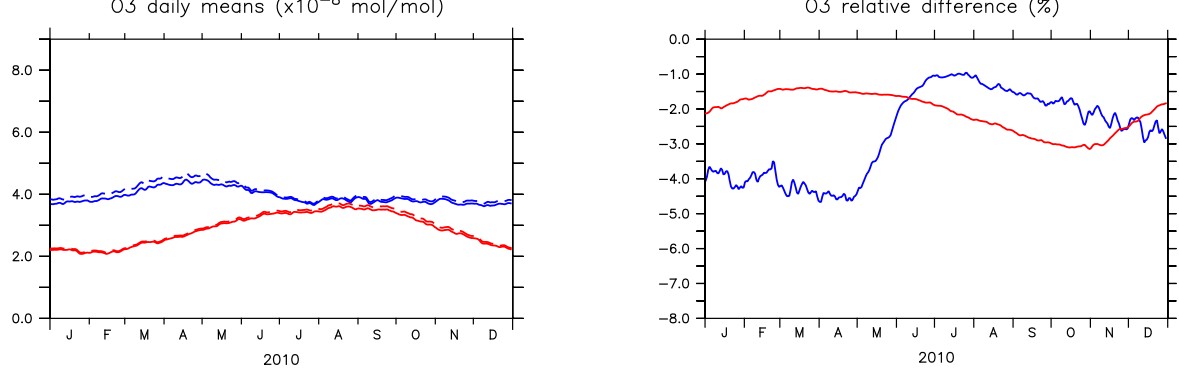

**Figure 7.** Same as in Fig. 4 for ozone (the unit in the left plot is $10^{-8}$ mol/mol.**((TO BE UPDATED))**

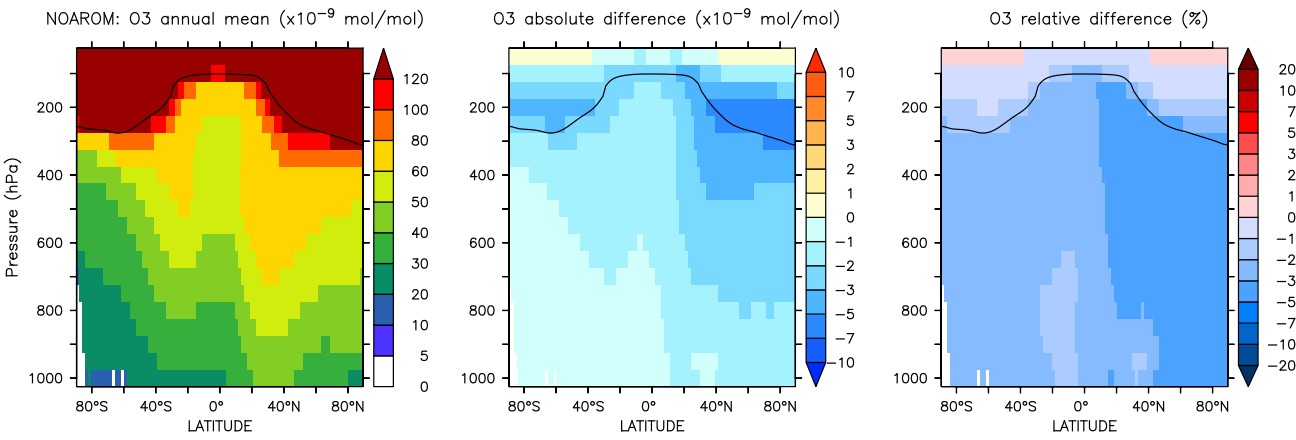

**Figure 8.** Annual average zonal mean O$_3$ mixing ratios. Left: Mixing ratios in the *NOAROM* simulation. Middle: Absolute difference *AROM-NOAROM*. Right: Relative difference *AROM/NOAROM*-1 in %. The solid line between 100 and 300 hPa depicts the mean tropopause level.

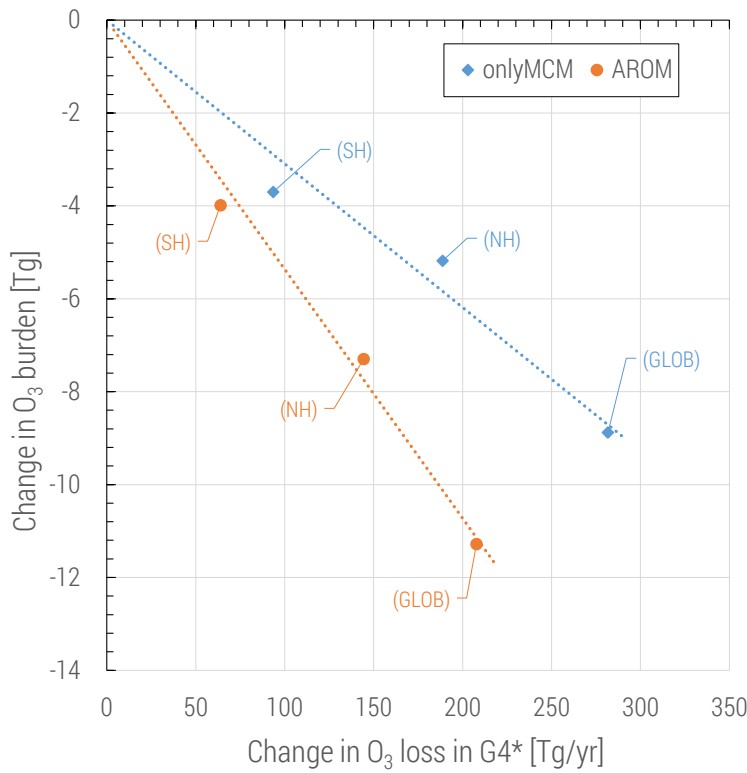

**Figure 9.** Change in tropospheric ozone burden versus change in ozone loss for all reactions in the VOC chemistry (G4 category of the MECCA mechanism[, see the Supplement of (Sander et al., 2019)]). The change in ozone loss is due to the reactions with (substituted) phenoxy radicals. Global and hemispheric results for *onlyMCM* (blue) and *AROM* (orange) simulations are shown.

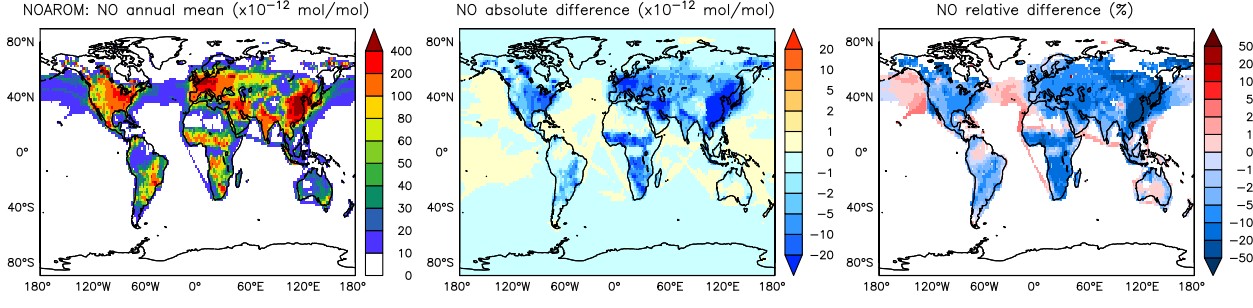

**Figure 10.** Annual average NO mixing ratios at the surface. Left: Mixing ratios in the *NOAROM* simulation. Middle: Absolute difference *AROM-NOAROM*. Right: Relative difference *AROM/NOAROM*-1 in % (shown only where NO is above 10 pmol/mol).

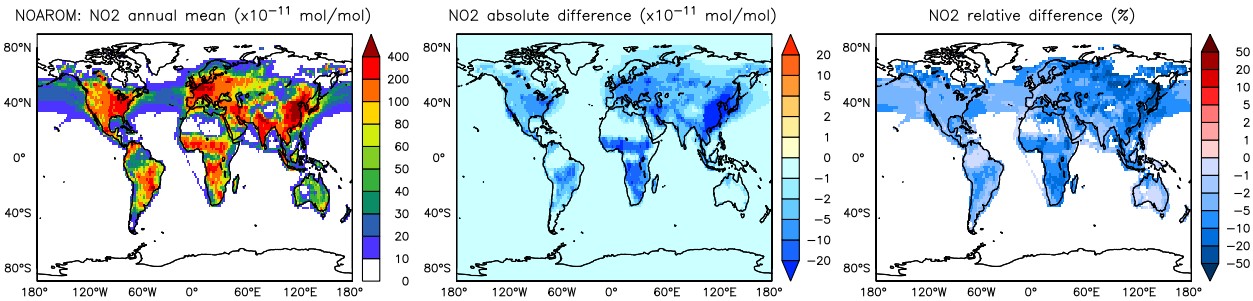

**Figure 11.** Annual average NO$_2$ mixing ratios at the surface. Left: Mixing ratios in the *NOAROM* simulation. Middle: Absolute difference *AROM-NOAROM*. Right: Relative difference *AROM/NOAROM*-1 in % (shown only where NO$_2$ is above 100 pmol/mol).

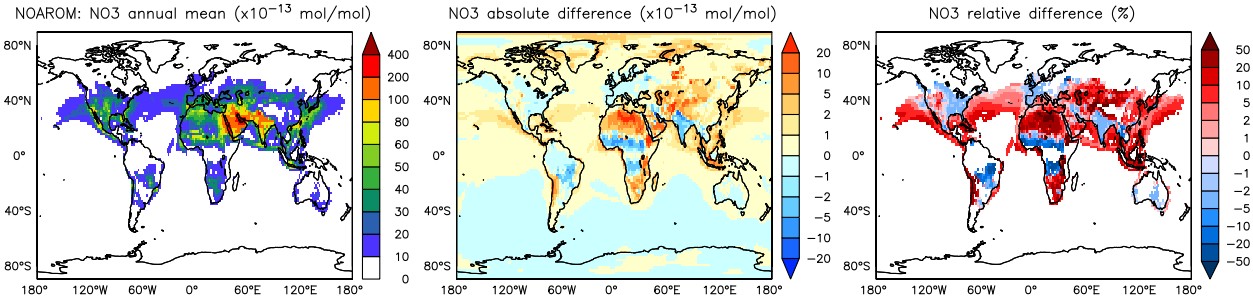

**Figure 12.** Annual average NO$_3$ mixing ratios at the surface. Left: Mixing ratios in the *NOAROM* simulation. Middle: Absolute difference *AROM-NOAROM*. Right: Relative difference *AROM/NOAROM*-1 in % (shown only where NO$_3$ is above 1 pmol/mol).

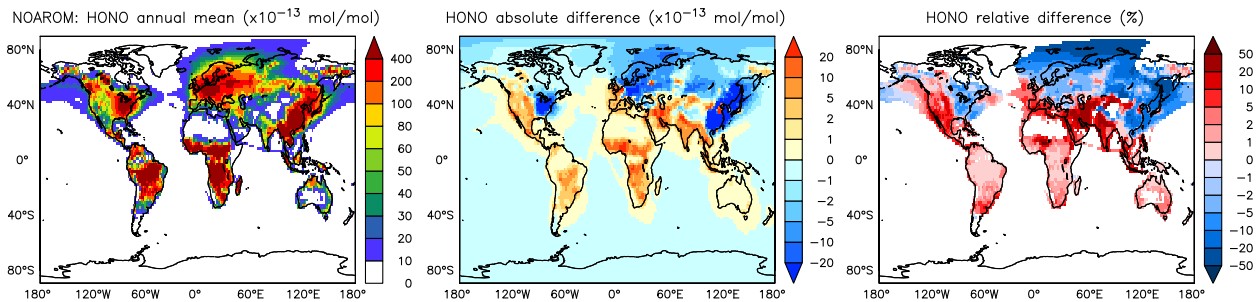

**Figure 13.** Annual average HONO mixing ratios at the surface. Left: Mixing ratios in the *NOAROM* simulation. Middle: Absolute difference *AROM-NOAROM*. Right: Relative difference *AROM/NOAROM*-1 in % (shown only where HONO is above 1 pmol/mol).

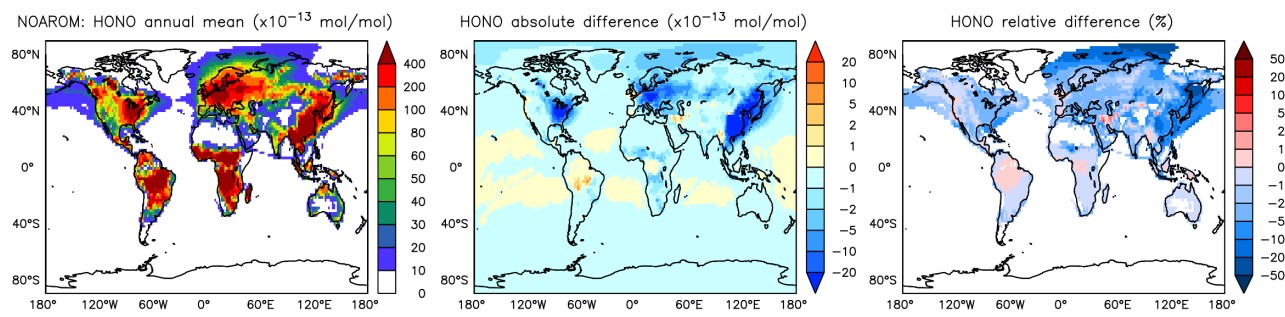

**Figure 14.** Annual average HONO mixing ratios at the surface. Left: Mixing ratios in the *NOAROM* simulation. Middle: Absolute difference *ONLYMCM-NOAROM*. Right: Relative difference *ONLYMCM/NOAROM*-1 in % (shown only where HONO is above 1 pmol/mol)

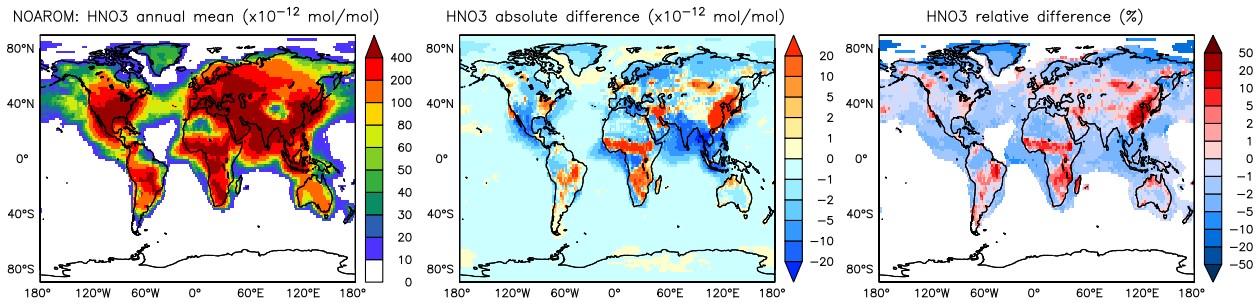

**Figure 15.** Annual average HNO₃ mixing ratios at the surface. Left: Mixing ratios in the *NOAROM* simulation. Middle: Absolute difference *AROM-NOAROM*. Right: Relative difference *AROM/NOAROM*-1 in % (shown only where HNO₃ is above 10 pmol/mol).

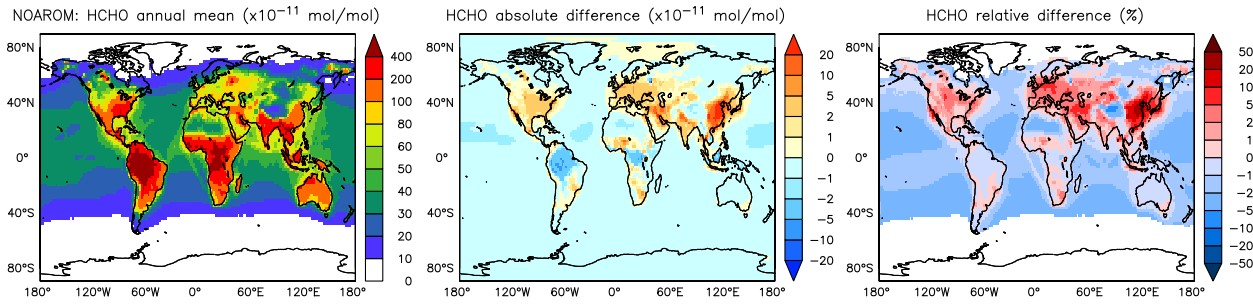

**Figure 16.** Annual average HCHO mixing ratios at the surface. Left: Mixing ratios in the *NOAROM* simulation. Middle: Absolute difference *AROM-NOAROM*. Right: Relative difference *AROM/NOAROM*-1 in % (shown only where HCHO is above 100 pmol/mol).

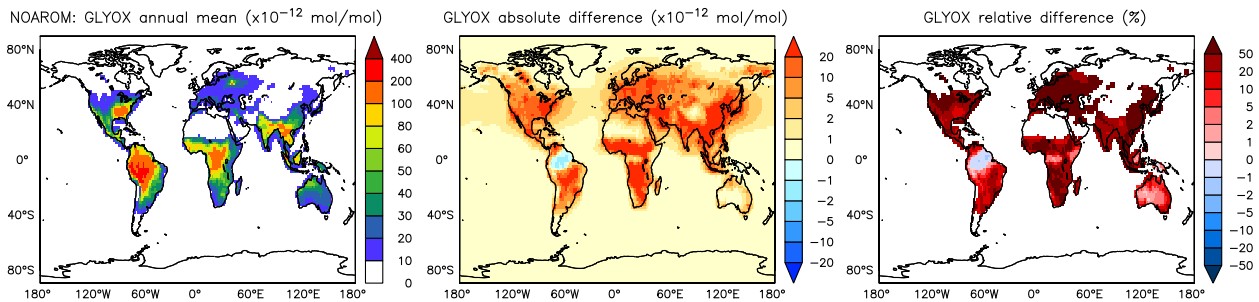

**Figure 17.** Annual average glyoxal mixing ratios at the surface. Left: Mixing ratios in the *NOAROM* simulation. Middle: Absolute difference *AROM-NOAROM*. Right: Relative difference *AROM/NOAROM*-1 in % (shown only where glyoxal is above 10 pmol/mol).

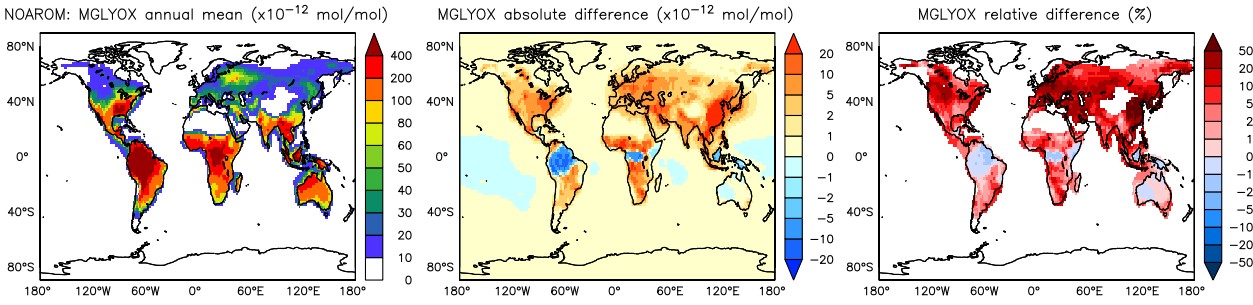

**Figure 18.** Annual average methyl glyoxal mixing ratios at the surface. Left: Mixing ratios in the *NOAROM* simulation. Middle: Absolute difference *AROM-NOAROM*. Right: Relative difference *AROM/NOAROM*-1 in % (shown only where methyl glyoxal is above 10 pmol/mol).

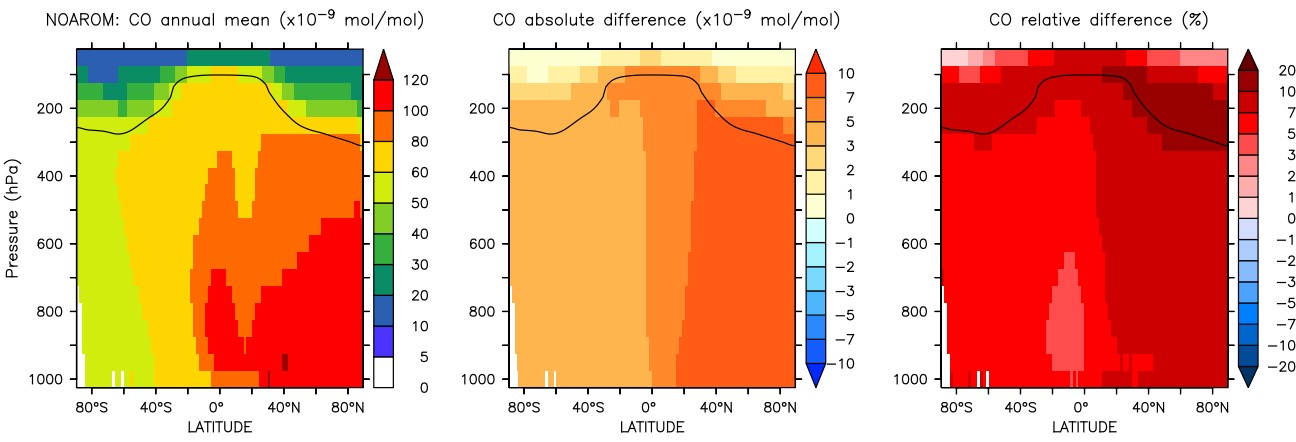

**Figure 19.** Annual average zonal mean CO mixing ratios. Left: Mixing ratios in the *NOAROM* simulation. Middle: Absolute difference *AROM-NOAROM*. Right: Relative difference *AROM/NOAROM*-1 in %. The solid line between 100 and 300 hPa depicts the mean tropopause level.

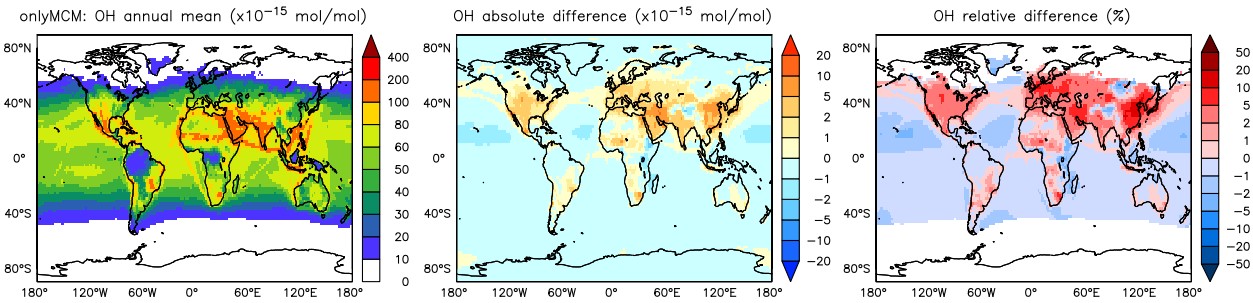

**Figure A1.** Annual average OH mixing ratios at the surface. Middle rows: Seasonal means. Left column: Mixing ratios in the *onlyMCM* simulation. Middle column: Absolute difference *AROM-onlyMCM*. Right column: Relative difference *AROM/onlyMCM*-1 in % (shown only where OH is above 0.01 pmol/mol).

**Appendix A: *AROM* vs. *onlyMCM***

In this appendix the impact of the modifications to the MCM chemistry (listed in Sect. 2) on the model results are shown for
the main atmospheric oxidants.

**Hydroxyl radical (OH)**

The differences at the surface are shown in Figure A1. Much of the increase in Figure 3 can be ascribed to the enhanced $HO_x$ production by photolysis of benzaldehyde (Roth et al., 2010) and HONO from R1. The latter from benzene chemistry explains the significant enhancement across the UT/LS (see Fig. A2).

**Ozone ($O_3$)**

The differences at the surface are shown in Figure A3. It can be seen that lareg part of the enhancement in surface ozone mixing ratio in Figure 6 is due to enhanced $HO_x$ production in regions that are not $NO_x$-limited. The zonal mean change in ozone is minimal and slightly positive at the tropical UT/LS (Fig. A4).

**Nitrate radical ($NO_3$)**

The differences at the surface are shown in Figure A5. It can be seen that the widespread enhancement of in Figure 12 is largely to be ascribed to the effect of phenylperoxy reaction with $NO_2$ (R2).

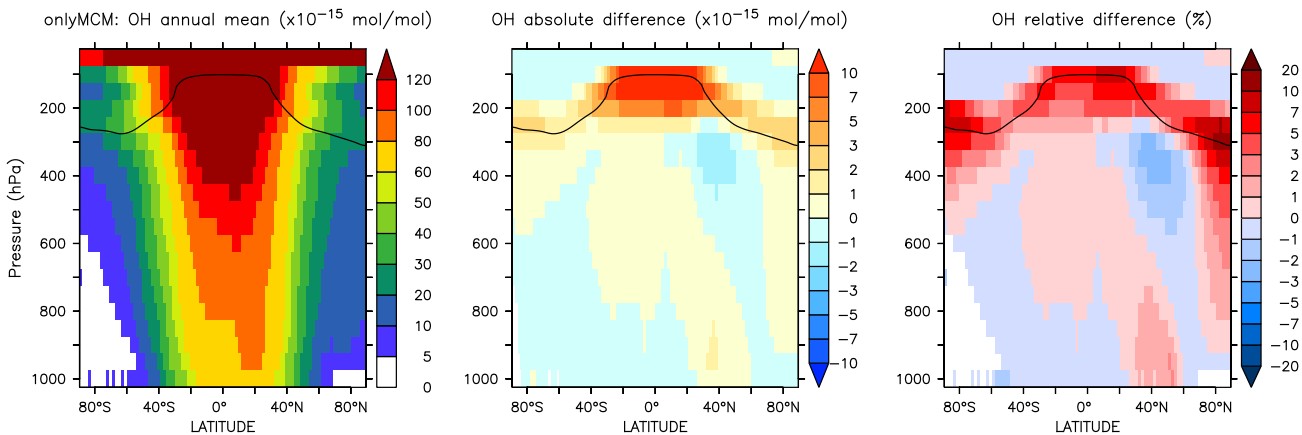

**Figure A2.** Annual average zonal mean OH mixing ratios. Left: Mixing ratios in the *onlyMCM* simulation. Middle: Absolute difference *AROM-onlyMCM*. Right: Relative difference *AROM/onlyMCM*-1 in %. The solid line between 100 and 300 hPa depicts the mean tropopause level.

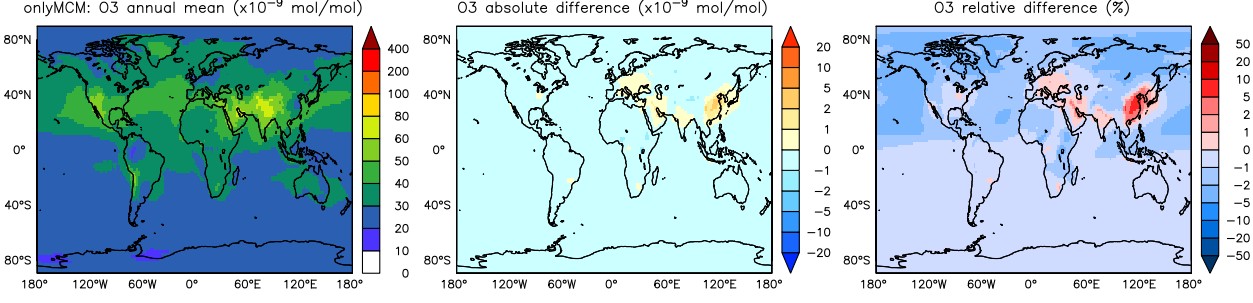

**Figure A3.** Annual average O₃ mixing ratios at the surface. Left: Mixing ratios in the *onlyMCM* simulation. Middle: Absolute difference *AROM-onlyMCM*. Right: Relative difference *AROM/onlyMCM*-1 in %.

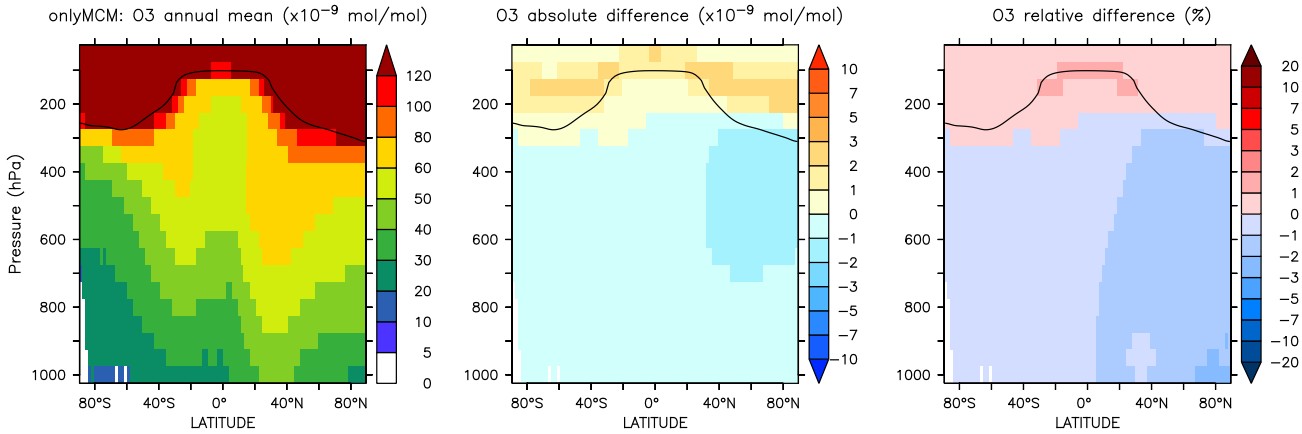

**Figure A4.** Annual average zonal mean $O_3$ mixing ratios. Left: Mixing ratios in the *onlyMCM* simulation. Middle: Absolute difference *AROM-onlyMCM*. Right: Relative difference *AROM/onlyMCM*-1 in %. The solid line between 100 and 300 hPa depicts the mean tropopause level.

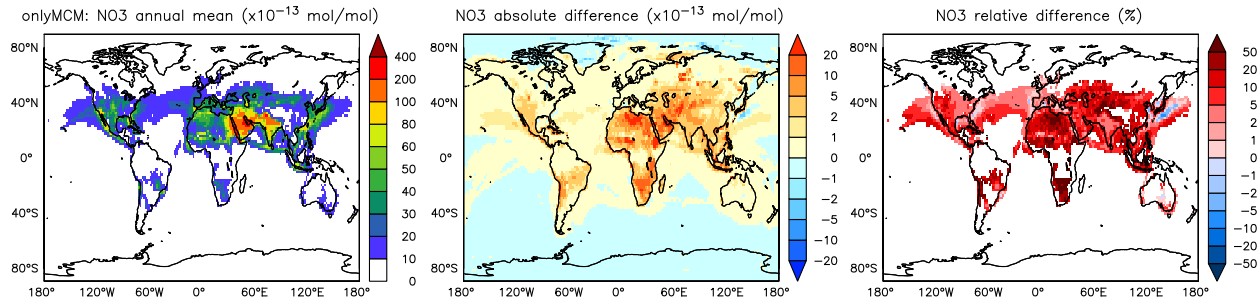

**Figure A5.** Annual average $NO_3$ mixing ratios at the surface. Left: Mixing ratios in the *onlyMCM* simulation. Middle: Absolute difference *AROM-onlyMCM*. Right: Relative difference *AROM/onlyMCM*-1 in % (shown only where $NO_3$ is above 1 pmol/mol).

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
