# Peer review of "Influence of aromatics on tropospheric gas-phase composition"

_Atmospheric Chemistry and Physics, 2020_

## Referee Comment (RC1) · Anonymous Referee #1 · 30 Jul 2020

Description:

This manuscript describes the changes in trace gas concentrations that occur when emissions of monocyclic aromatic compounds are included in a global general circulation / atmospheric chemistry model. Many recent studies have pointed out the strong influence of aromatic compounds on local and regional air quality, especially in East Asia, which makes this manuscript a timely assessment of their treatment in models and their global impacts. Here, the authors incorporate the MECCA mechanism into EMAC and diagnose the implications of aromatic chemistry by comparing a simulation with aromatic emissions turned on to one with aromatic emissions turned off. Unsurprisingly, they find that aromatic compounds contribute substantially to the global budgets of glyoxal and methylglyoxal – two of their main oxidation products and important

SOA precursors – while their contributions to the budgets of smaller and more ubiquitous carbon-containing compounds, such as HCHO and CO, are smaller relative to the background. For budgets of HOx, NOx, and ozone, the effects are more complex; while aromatic emissions have small global effects (which, intriguingly, sometimes contradict those found by previous studies), their local effects can be moderately substantial and can vary in sign depending on local chemistry.

General comments:

The manuscript is straightforwardly written, well laid-out, and presents its findings clearly. Furthermore, because the authors use a highly comprehensive mechanism based on MCM, the chemical outcomes of aromatic oxidation are likely more robust than previous studies. However, it appears from looking more closely at the mechanism that some of the larger aromatic species (e.g. xylene and ethylbenzene) are not actually treated explicitly, but are oxidized in pathways identical to those of toluene with a "LCARBON" counter species denoting the carbon imbalance that arises from this treatment. This should be more explicitly described in the manuscript, which appears to imply that all aromatic species are treated independently by providing their individual emissions.

Another general concern is that in many places model outcomes are simply described without sufficient discussion of their causal pathways, which strongly diminishes the usefulness of these findings. Further, many of the important effects of these model outcomes are only touched on without any quantitative discussion – e.g., the changes to methane lifetime and SOA formation from glyoxal and methylglyoxal. Specific instances are pointed out in the comments below.

Finally, while the short discussion of uncertainties in Section 4 is a particularly useful addition to this manuscript, it does not go far enough to provide the reader with either quantitative or qualitative bounds on the model outcomes described herein. Of particular note, more attention should be paid to: (a) the effects of grid resolution –

given that there's a lot of spatial heterogeneity in model outcomes, and the effects even change sign depending on local conditions, is 1.875ËŽÂ̆square enough to resolve this chemistry? What outcomes might be masked by the artificial mixing that occurs in such large-scale grid boxes? (b) uncertainties in chemical mechanisms, especially the newly implemented ones described in the bullet point list in Section 2 – while there is some discussion in Section 4 of the uncertainty in the phenoxy + O3 reaction, it should be put into a larger context and more explicitly tied to the uncertainty bounds of model outcomes that might be expected given the uncertainties in the chemical mechanism; and lastly (c) uncertainties in emissions – while these are also discussed briefly in Section 4, the scope of the discussion is very limited and does not leave readers with any quantitative understanding of how well the emission totals are known, what their overall interannual variability might be, and how this could affect the model outcomes described in Section 3. I am not trying to argue that the authors need to perform additional sensitivity simulations, but the uncertainties merit a more lengthy, detailed, and quantitative description than is provided here. Additional (more specific) questions about uncertainty are given below.

Specific comments:

L 19 - Does the 200-300 Tg/yr refer to Tg O3?

L 69-72 - Because the primary findings of this paper rely so heavily on the magnitude of these emissions, some small discussion of their uncertainty is warranted. Do the sources from which the inventories were derived describe the range of plausible emission amounts? Do different anthropogenic or biomass burning inventories give different emission amounts? EDIT - I see this is partially addressed in Section 4 (though it would be useful here to direct the reader to the later discussion of uncertainties). However, the short paragraph about emissions uncertainties still lacks quantitative detail, and most of the questions above still remain unanswered. Also, can a numerical range of inter-annual variability of total pyrogenic aromatic emissions be provided?

[Figure]

L 75 - It would be helpful here to give detail not only on the additions that have been to the mechanism, but also on generally what simplifications were made to MCM to arrive at MECCA. I realize that's available in Cabrera-Perez et al. 2016, but the reader should be able to get a sense of the methodology here without having to fully read another paper. In particular, it is problematic that this manuscript implies a full detailed mechanism for the suite of aromatics shown in Table 1 when in fact many of them just use the same oxidation schemes.

L 122-124 - This is not a complete explanation of the OH increase in high-NOx regions. It is stated that the increased OH is "caused by the reaction of NO with HO2", but it was also stated two lines previously that NO decreases in these regions. Does HO2 increase enough as to offset both the NO decrease and the OH decrease through direct reaction with aromatics? What causes such a pronounced HO2 increase? Further, Figures 4 and 5 are not particularly useful to the reader without an explanation of why these effects occur. What causes the seasonal variability in the effects of aromatics on OH? Why are they strongest in the upper troposphere?

L 123 - Is this "positive correlation" a quantitative effect, diagnosed by some sort of regression analysis (across time? or just regions?), and if so, can it be explained in more detail here? If this "positive correlation" is just meant to say that including anthropogenic emissions causes an increase in OH, then this phrasing is misleading – better just to state simply that OH is higher in AROM than NOAROM in these regions. Also, are these correlations really diagnosed from anthropogenic emissions in particular, or from the inclusion of all aromatic emissions (including biogenic and pyrogenic)?

L 132 - The increased methane lifetime is likely to be of great interest to readers. Can it be quantified?

L 138-140 - The same comment above (L 123) applies here to the "correlation" phrasing.

L 141 - It appears Figure 9 is discussed here in the text before Figures 7 (L 143) and 8

(L 157)

L 141-145 - The same comment above (L 122-124) applies here; why do these seasonal and zonal patterns arise?

L 153 - Can some more quantitative description of the variation with tropopause definition (and a description of the definition itself) be included here? What specifically is meant by "robust"? Figure 7 makes it look instead like there are large absolute differences right at the tropopause, which would imply that the definition might be highly important.

L 154 - What is meant by "these changes", and why does the explanation provided here differ from those above (L 134-136)?

L 159 - Extra close-parentheses.

L 143-164 - This paragraph is long and covers a wide range of different topics; consider splitting it up? Also, the sentences around L 156-160 seem to be in an odd order; the sentence beginning "However, with aromatics" seems like it should be followed by the sentence beginning "Ozone is known to react", while the two intervening sentences seem like a non sequitur.

L 161-164 - How well is this chemistry known? The cited study describes the reactions of the phenoxy radical with O3 and with other phenoxy radicals, but presumably this is also in competition with many other reactions, including possible unimolecular rearrangements or decomposition. Have competitive studies been reported? If not, can some estimate of competing reactions rates be used along with the uncertainty bounds of the phenoxy + ozone reaction rate to determine some level of certainty for this discussion here? Considering how strong the simulated effect is, and how its catalytic nature under some conditions could magnify even small errors, some discussion of uncertainty is warranted. EDIT - I see this is partially addressed below in section 4 (it'd be nice to have some parenthetical here directing the reader to the later discussion on

uncertainties). Some questions remain, though – most notably why the rate constant "has to be regarded as a lower limit" (L 227) and whether competing reactions might also be uncertain.

L 164 - No period (assuming this is the end of the sentence).

L 173-174 - How does nitrophenol photolysis come into play here for the phenoxy radicals? R1 implies either that nitrophenol photolysis does not form phenoxy radicals or that the mechanism skips the phenoxy radical step and goes directly to decomposition products.

L 186-187 - This phrasing does not make it particularly clear which simulation has the higher NO3 concentrations.

L 188 - Why does this effect occur in places with pyrogenic aromatic emissions, while the HOx and O3 effects described previously are more strongly associated with anthropogenic emissions? The same question applies to the HONO increase on L 191.

L 198-200 - What is the explanation for these findings?

L 208-209 - Was the effect on the SOA budget quantified here? It seems this effect would be of great interest to some readers.

L 218 - It seems odd that so much detail is shown in the many other figures for other species discussed in this manuscript, but this interesting finding for CO is not shown. Can a CO figure be added, and can an explanation be given for these spatial effects?

L 230-231 - It was earlier implied (L 173-174) that the model *does* account for the effects of photolysis of nitrophenols. Can this be clarified?

Table 5 is cut off by the end of the page.

Can the OH mixing ratios in Figures 3-5 be expressed instead in the more commonly used concentration units?

[Figure]

Figure 4 is very confusing. What are the units on the left plot? Why are there two very close blue and red lines in the left plot? Why does the caption reference solid and dashed lines in the right plot when only solid lines exist? The same questions apply to Figure 9.

---

## Referee Comment (RC2) · Alexander Archibald (Referee) · 4 Aug 2020

Taraborrelli et al. provide an updated mechanism for the oxidation of aromatics in the EMAC model and a summary of the impacts of this update on key trace gases in the troposphere.

This is a generally well written and well executed study but I have several minor comments in the attached pdf and a few more major comments before recommending publication.

Major comments:

Comparison to observations is lacking which limits the sense I get that the changes are in anyway in the right direction. For example, the changes in surface ozone and NO2 in

EAS are large and I would imagine significant. It would be good to see how these compare with observations. Whilst I agree with the previous reviewers comments about model resolution and whilst there may well be structural errors in EMAC that mean that even with a better representation of the chemistry the comparison to observations is worse, I feel some comparison to observations is warranted. I also feel this will help focus the paper as currently it reads as one in which aromatics vs no-aromatics is the focus, but given we know aromatics are important (and abundant in urban environments) shouldn't the focus be Arom vs OnlyMCM? At least I find this comparison more interesting than Arom vs NoArom.

It would be good if there were some figures (perhaps in an appendix) which compare the OnlyMCM and Arom scheme under idealised (Box model) conditions. Ideally this would be against laboratory data but I think even against some general scenarios it would be very useful to see how the differences implemented affect the results and then some sensitivity analysis could be performed I think quite straightforwardly to look at the impacts of some of these uncertain thermal-kinetic and photolytic processes.

Please also note the supplement to this comment:
https://acp.copernicus.org/preprints/acp-2020-461/acp-2020-461-RC2-supplement.pdf

**Supplement:**

[Figure]

[Figure]

**Influence of aromatics on tropospheric gas-phase composition**

[revised manuscript text omitted]

- For several nitrophenols (MCM names: HOC6H4NO2, DNPHEN, TOL1OHNO2, MNCATECH, DNCRES), their photolysis reactions were added (Bejan et al., 2006), e.g.: L78: How were they added? i.e. what cross-sections and quantum yields used?

$$\text{OH, NO}_2 \xrightarrow{h\nu} \text{O} \diagdown \diagup \diagdown \text{O} + \text{HONO} + \text{CO}_2 + \text{CO} + 2\,\text{HO}_2 \quad \text{(R1)}$$

80 - For the photolysis of benzaldehyde, the MCM uses the rate constant ($j$-value) of methacrolein as a proxy. We have calculated the $j$-value based on the UV/VIS spectrum of benzaldehyde recommended by Wallington et al. (2018). In our code, the photolysis of benzaldehyde produces $\mathrm{C_6H_5O_2}$, $\mathrm{HO_2}$ and CO.

[Figure]

[Figure]

- For several phenyl peroxy compounds (MCM names: C6H5O2, CATEC1O2, OXYL1O2, MCATEC1O2, NCRES1O2), their reactions with $NO_2$ were added (Jagiella and Zabel, 2007), e.g.:

85

(R2)

- For the reaction of $HO_2$ with the peroxy radical C6H5CO3 (resulting from the oxidation of benzaldehyde), we use the yields provided by Roth et al. (2010).

- Alkyl nitrate yields are calculated as a function of temperature and pressure, as described by Sander et al. (2019).

- Bicyclic peroxy radicals in the oxidation mechanism of toluene produce some glyoxal and methyl glyoxal as suggested

90  by Birdsall et al. (2010). Benzene is treated analogously.                L89: Please be quantitative.

The aerosol calculations follow the approach of Pringle et al. (2010), with the notable difference of the inclusion of the explicit organic aerosol submodel ORACLEv1.0 by Tsimpidi et al. (2014). Although, similar to Tsimpidi et al. (2014), low- and intermediate volatiles are parameterized as lumped species, the equilibrium with their equivalent aerosol phase is explicitly calculated for $\simeq 600$ volatile organic carbon tracers via ORACLE. The volatility and the enthalpy of vaporization of each tracer

95  is estimated with the approaches of Li et al. (2016) and Epstein et al. (2010), respectively.

The simulated period covers the years 2009–2010, with the first year as spin-up, and the year 2010 being used for the analysis. The feedback between radiation and chemistry was decoupled to avoid any influence of chemistry on the dynamics (QCTM mode by Deckert et al. (2011)). As a consequence, every simulation discussed here has the same meteorology, i.e., binary identical transport.

100  To analyze the influence of the aromatic compounds on atmospheric chemistry and composition, we performed three model simulations, as listed in Tab. 2. The *AROM* simulation includes all chemical reactions and emissions of the following mono-cyclic aromatic compounds: benzene, toluene, xylenes (lumped), phenol, styrene, ethylbenzene, trimethylbenzenes (lumped), benzaldehydes, and higher aromatics (as representative of aromatics with more than 9 carbon atoms). The reference simula-tion (*NOAROM*) is identical to *AROM*, except that it excludes aromatic compounds. In the *ONLYMCM* run, we reverted the

105  additions and changes to the MCM that have been described above. Our focus is to compare *AROM* with *NOAROM*. Results of *ONLYMCM* are mainly interesting for benzaldehyde and HONO. As EMAC uses terrain-following vertical hybrid pressure coordinates, we will refer to "surface" as the lowest model level, with an average thickness of roughly 60 m.

[Figure]

**3 Results and discussion**

L110: Please plot the data in nmol/mol to make things clearer for the reader.

Globally averaged surface mixing ratios obtained from all model simulations (*AROM*, *NOAROM*, and *ONLYMCM*) are listed in

110  Tab. 3. Figure 1 shows the annual average mixing ratios of the sum of all aromatic compounds included in the simulation *AROM*. They are higher in continental areas and close to the surface. The highest values are predicted in the northern hemisphere (NH), in particular, in East and South Asia, as well as in parts of Europe, Africa, and the US, reaching up to about 1 nmol/mol. The background mean mixing ratios in oceanic areas of the southern hemisphere (SH) are of the order of a few pmol/mol. For a more detailed analysis, we have selected the following five regions, as defined in Figure 2: Amazon area (AMA), central Africa

115  (CAF), eastern Asia (EAS), Europe (EUR), and eastern US (EUS). The budgets of selected chemical species were calculated within these regions (Tab. 5).

**3.1 Hydroxyl radical (OH)**

L118: Insert "surface" between these two words.

Figure 3 shows the model-calculated OH in the *AROM* and *NOAROM* simulations. When aromatics are introduced to the model, the global average concentration of OH decreases for two reasons: first, the direct reaction with aromatics consumes

120  OH, and second, additional CO resulting from the degradation of aromatics represents an increased sink for OH. However, in eastern Asia, Europe, and the east coast of the US, where $NO_x$ concentrations are high, an increase of OH can be seen. Although the aromatics decrease $NO_x$ in these areas (see below), the chemical system remains in the high-$NO_x$ regime.

We find a positive correlation between OH and anthropogenic emissions in these regions but a negative correlation in the low-$NO_x$ CAF region. The increased OH in the high-$NO_x$ regions is mainly caused by the reaction of NO with $HO_2$.

125  Figure 4 shows the seasonal cycle of the OH mixing ratio in the planetary boundary layer for the NH and SH. Inclusion of the aromatics leads to a relative decrease between 2.5 % and 5.5 %. Higher OH concentrations are identified over continental areas during the NH autumn, winter and spring than in summer (Fig. 3). In summer, OH concentrations increase only at a few locations when aromatics are included.

Figure 5 shows the annual zonal mean changes of the OH mixing ratio. The changes are most pronounced in the NH

130  upper troposphere where reductions range from 7 % to 20 %. This helps bringing the model-simulated inter-hemispheric OH asymmetry closer to that derived from observations (Lelieveld et al., 2016). Globally, aromatics oxidation reduces OH by 7.7 % and consequently increases methane lifetime.

L131-132: Can you be more specific on both the impact on the OH NH:SH ratio change and the impact on the methane lifetime.

**3.2 Ozone ($O_3$)**

In most areas of the globe, surface ozone is slightly lower in *AROM* than in *NOAROM* (Fig. 6). The $O_3$ reduction is due

135  to (i) the decrease in $NO_x$ concentrations (limiting ozone formation) and (ii) increasing radical production (OH, $HO_2$, and $RO_2$) in ozone-depleting regimes, which enhances reactions of $O_3$ with $HO_2$ and OH. Only a few high-$NO_x$ regions, where hydrocarbons are the limiting factor for ozone formation, show increased ozone concentrations: mainly East China (EAS), but also the eastern US (EUS) and Europe (EUR). The increases in these areas correlate with anthropogenic emissions of

L135-136: Has there been an increase in the flux through O3+OH? I'm surprised given the OH has gone down in these regions.

[Figure]

[Figure]

L153: Can you confirm which definition you used in the analysis?

L141: Odd to ref. Fig 9 before 7 or 8. Re-order?

aromatics, which have significant ozone formation potentials. We find a positive correlation between $O_3$ and anthropogenic

140   emissions in the EAS and EUR regions but a negative correlation in the low-$NO_x$ CAF region.

The seasonal cycles of the relative differences show lower amplitude than for OH, but similar patterns (Fig. 9). The impact of aromatics is smallest in summer.

The zonal mean changes of $O_3$ mixing ratio in the troposphere are uniformly negative (Fig. 7). Similar to surface ozone, the annual mean changes for *ONLYMCM* and *AROM* are $-2.3\%$ and $-3.0\%$, respectively. The hemispheric changes are shown

145   in Tab. 4. It is well known that MCM for aromatics overestimates ozone production in chamber experiments (Bloss et al., 2005b). The issue has been analysed in the companion paper (Bloss et al., 2005a) where the best mechanism improvement was found to be an early OH source during oxidation. Cabrera-Perez et al. (2016) introduced enhanced $HO_x$-sources by photolysis of benzaldehyde and nitrophenols. These modifications consistently result in less ozone produced with respect to MCM. These results deviate from the results by Yan et al. (2019) who suggested a global increase of 0.4 % due to aromatics.

150   However, they only considered benzene, toluene and xylenes. Our results, obtained with a more comprehensive setup, suggest that aromatics could slightly ameliorate the model overestimate in the NH (Jöckel et al., 2016; Young et al., 2018). The overall tropospheric ozone burden decreases from 381 to 369 Tg for the *AROM* simulation. These estimated changes are robust against the tropopause definition and are about -3.5 and -2.3 % for the Northern and Southern Hemispheres, respectively (Table 4). These changes are associated with the enhanced direct ozone loss by reactions with organic compounds. It is widely

155   acknowledged that this direct loss is only due to the ozonolysis of unsaturated VOCs and is estimated to be about 100 Tg/yr, less than 2 % of the tropospheric ozone budget (Tilmes et al., 2016). However, with aromatics a new direct ozone loss process involving organic radicals comes in place. In Figure 8 the change in tropospheric ozone burden is shown against the change in ozone loss with organic compounds. This change is estimated to be globally in the 200-300 Tg/yr range depending on the mechanism used and is comparable to the loss by bromine chemistry in the troposphere (Sherwen et al., 2016)). Ozone is

160   known to react with organic radicals like methyl peroxy radical although this loss is an insignificant sink (Tyndall et al., 1998). We find that phenoxy radicals from aromatics are a significant sink term of ozone ($>200$ Tg/yr). These radicals are unique to aromatics oxidation and they also react with NO and $NO_2$. When the concentrations of $NO_x$ are relatively low, $C_6H_5O$ has sufficiently long lifetime to react with $O_3$. This ozone loss is modelled based on the results by Tao and Li (1999) for phenoxy radical

165

L155: Is it widely acknowledged that it is "only" ozonolysis? And does the definition of loss change with different constructs of the O3 budget (c.f. Bates and Jacob 2019)?

(R3)

[Figure]

This ozone loss is enhanced by phenoxy radical production in the R2 reaction and the subsequent loss of odd oxygen by

$NO_3$ photolysis and $N_2O_5$ heterogeneous loss

L166-167: Can you quantify the relative contribution of these different pathways to the 200 Tg/yr O3 (odd oxygen?) loss?

$$NO_3 + h\nu \quad \rightarrow \quad NO + O_2 \tag{R4}$$

$$NO_3 + NO_2 \quad \rightarrow \quad N_2O_5 \tag{R5}$$

170 $$N_2O_5 + H_2O \quad \rightarrow \quad 2HNO_3(aq) \tag{R6}$$

L174: But the way you have written R1 suggests that phenoxy radicals are not formed (instead butenedial is formed).

[revised manuscript text omitted]

---

## Author Comment (AC2) · 16 Dec 2020

**Reply to RC2 on acp-2020-461**

Taraborrelli *et al.*

December 16, 2020

Dear Alexander Archibald (Referee #2),

thank you for taking the time to review of our manuscript. It gave us the chance to improve the manuscript significantly.
Please find below a point-by-point response to your comments.

> **Description**
> *Taraborrelli et al. provide an updated mechanism for the oxidation of aromatics in the EMAC model and a summary of the impacts of this update on key trace gases in the troposphere.This is a generally well written and well executed study but I have several minor comments in the attached pdf and a few more major comments before recommending publication.*

> **Major comments**
> *Comparison to observations is lacking which limits the sense I get that the changes are in anyway in the right direction. For example, the changes in surface ozone and NO2 in EAS are large and I would imagine significant. It would be good to see how these compare with observations. Whilst I agree with the previous reviewers comments about model resolution and whilst there may well be structural errors in EMAC that mean that even with a better representation of the chemistry the comparison to observations is worse, I feel some comparison to observations is warranted.*

Reply:
We agree with the referee that comparison with observations would strengthen and focus the manuscript. However, a comprehensive evaluation of the EMAC model with the complex organic chemistry (MOM) and against satellite retrievals of $O_3$ and $NO_2$ is in preparation and will be soon submitted for peer review. A first comparison of the model results with IASI-FORLI retrievals for ozone along with a detailed $O_x$ budget is currently presented in Rosanka et al. (2020). The model results for tropospheric ozone, with the modified MCM chemistry of aromatics we present here, are clearly still too high with overestimates of up to 10 DU. This positive bias will be addressed by further improving existent parametrizations in EMAC like the dry deposition scheme (Emmerichs et al., 2020) and extending the representation of multiphase chemistry, which started with Rosanka et al. (2020), to deliquescent aerosols.

> *I also feel this will help focus the paper as currently it reads as one in which aromatics vs no-aromatics is the focus, but given we know aromatics are important (and abundant in urban environments) shouldn't the focus be Arom vs OnlyMCM? At least I find this comparison more interesting than Arom vs NoArom.It would be good if there were some figures (perhaps in an appendix) which compare the OnlyMCM and Arom scheme under idealised (Box model) conditions. Ideally this would be against laboratory data but I think even against some general scenarios it would be very useful to see how the differences implemented affect the results and then some sensitivity analysis could be performed I think quite straightforwardly to look at the impacts of some of these uncertain thermal-kinetic and photolytic processes.*

Reply:

We thank the referee for this comment. Currently, we are not setup to compare box model simulations to lab data from chamber experiments. Our *modus operandi* is to obtain an intermediate and faithful reduction of a chemical mechanism like MCM that has been widely used and tested against lab measurements. We also believe that the differences in the results between AROM and onlyMCM are interesting to show. Instead of showing box model simulation results under idealized conditions, we think that showing the global distribution of the differences (spanning many possible scenarios) between AROM and onlyMCM simulations is a synthetic and useful way to visualize the deviations across a comprehensive set of chemical regimes. We therefore added an appendix to the manuscript to enhance the discussion of the differences between the MCM mechanism and our mechanism for the aromatics. In this appendix we briefly present the differences for the main oxidants OH, $O_3$ and $NO_3$.

**Specific Comments**

*L13: Changes of what?*

Reply:

We added "of trace gas levels" to the text.

*L19: Is this a net loss? If not, is it a very important finding?*

Reply:

This is a direct loss of ozone. As it is localized in the upper troposphere where benzene is transported and NO levels are generally low, this turns into a net loss of ozone. However, model setup used in this study did not have a comprehensive set of passive tracers that allows the classical tropospheric $O_x$ budget to be computed. Nevertheless, we think that this result, the direct loss of ozone, is worth noting especially because it is clearly missing in all other global models with which the global impact of aromatics on ozone has been estimated to be positive, contrary to our study.

*L31: This is too vague. There are specific definitions of aromaticity with implications for the chemistry of compounds in this class.*

Reply:

We agree with the referee that we need to be more specific in this respect. Thus, we have changed the first sentence of the paragraph by stating that aromatics are unsaturated planar cyclic organic compounds with enhanced stability due to a strong electron delocalization.

*L38: Add a reference for the toluene biogenic emissions.*

Reply:

We added the reference to the first reported biogenic emission of toluene by Heiden et al. (1999).

*L42: Is that true of all aromatics? i.e. benzene?*

Reply:

We agree with the referee that it might sound odd to put benzene in the category of organic compounds that have a high reactivity. We have changed the relative sentence by removing the reference to the high reactivity and expressed in more neutral terms with a range of tropospheric lifetimes.

*L46-50: There is a rich literature on many aspects of this chemistry which should be cited.*

Reply:
We agree with the referee and we added the references to the review papers by Atkinson and Arey (2003) and Vereecken (2019). For the SOA formation from aromatics oxidation we now refer to Henze et al. (2008) and Lin et al. (2012).

*L78: How were they added? i.e. what cross-sections and quantum yields used?*

Reply:
We apologize for the lack of detail here. For the photolytic HONO-formation from nitrophenols the cross sections and quantum yield provided by Chen et al. (2011) are used by the JVAL and JVPP models (Sander et al., 2014) for calculating the $j$-values. We have modified the manuscript accordingly.

*L89: Please be quantitative.*

Reply:
We have added to the revised manuscript the information on the yields of glyoxal (60%) and methyl-glyoxal (40%) for toluene from Birdsall et al. (2010). We also specify now that these yields are for the non-radical terminating channels in the reactions with NO and $HO_2$.

*L110: Please plot the data in nmol/mol to make things clearer for the reader.*

Reply:
Yes, we now plot the data with mol/mol and the appropriate exponent for the range of values shown. We agree it was not clear before.

*L118: Insert "surface" between these two words.*

Reply:
Done.

*L131-132: Can you be more specific on both the impact on the OH NH:SH ratio change and the impact on the methane lifetime.*

Reply:
Referee #1 had a similar comment and we acknowledge that the quantification of the impact on methane lifetime could have been given more space than a short mention without referring to Table 4. Therefore, we have extended the paragraph L129-132 by pointing explicitly to Table 4 and shortly discussing the changes in OH and CH4 lifetime in the two hemispheres.

*L135-136: Has there been an increase in the flux through O3+OH? I'm surprised given the OH has gone down in these regions.*

Reply:
We thank the referee for spotting this inconsistency. Clearly, the simulation results do not support the statement on an increase in the flux of the $O_3$ + OH reaction in ozone-depleting regimes, e.g. over the ocean. We have removed OH from this explanation.

*L141: Odd to ref. Fig 9 before 7 or 8. Re-order?*

Reply:
Thank you for spotting this. The figures are reordered now.

*L153: Can you confirm which definition you used in the analysis?*

Reply:
Tropospheric burdens were reckoned using six different tropopause definitions (provided by the TROPOP submodel, see Jöckel et al. (2010) for details): 1,2) surfaces of $O_3$ mixing ratio of 125 and 150 nmol/mol, respectively, 3) WMO definition (WMO (1957)), 4) dynamic PV-based (3.5 PVU potential vorticity surface, sought within 50–800 hPa), 5) climatological (invariable zonal profile, i.e. $300\text{-}215\times(\cos(\text{latitude}))^2$ hPa) and 6) the combined definition (WMO tropopause within 30°N–30°S, otherwise dynamic PV-based tropopause). The latter definition is used by default in EMAC and in this manuscript to report tropospheric budgets. Estimated changes to tropospheric $O_3$ burden are identical within 0.05% between the available definitions, which we conclude as robust against the definition used. We now put this information in the caption of Table 4.

*L155: Is it widely acknowledged that it is "only" ozonolysis? And does the definition of loss change with different constructs of the O3 budget (c.f. Bates and Jacob 2019)?*

Reply:
According to the expanded definition of the odd oxygen budget by Bates and Jacob (2020), the loss ozone from reaction with phenoxy radical would count as half since $RO_2$ formation is counted with the "stoichiometric" coefficient 0.5 in the $O_y$ family. This coefficient is justified in order to account for the effect the $O(^1D) + H_2O$ reaction has on OH. However, the rationale and validity of this "stoichiometric" accounting for peroxy radicals is not clear. We agree that when ozone reacts with phenoxy radical close to the pollution sources the NO-to-$NO_2$ conversion by the resulting phenyl peroxy radical would largely compensate the ozone loss in question. However, in the upper troposphere where benzene is transported and where NO levels are usually low, the loss of ozone with phenoxy is a net loss.

*L156: Perhaps add e.g., as this is just one models calculation.*

Reply:
Done.

*L161: Confirm if you mean ozone or odd-oxygen?*

Reply:
We mean ozone.

*L166-167: Can you quantify the relative contribution of these different pathways to the 200 Tg/yr O3 (odd oxygen?) loss?*

Reply:
We now realize that our formulations have been not clear and misleading. The 200-300 Tg/yr we give in the manuscript is the direct ozone loss in the reaction with (substituted) phenoxy radicals. What we wanted to express here was that the phenyl peroxy radical produced by reaction R3 enhances the $NO_3$ formation at night, which in turn enhances the $O_x$ via the heterogeneous loss of $N_2O_5$. Having no detailed passive tracers for computing the $O_x$ budget in this study, we cannot quantify the strength of the $O_x$ destruction we describe. We now make this clearer in the revised manuscript.

*L174: But the way you have written R1 suggests that phenoxy radicals are not formed (instead butenedial is formed).*

Reply:
Indeed R1 destroys the aromaticity of the molecule and therefore any possibility to form further (substituted) phenoxy radicals. Unfortunately, we have not explicitly mentioned that in MCM (AROM and onlyMCM simulations) the reactions of the simplest nitrophenol (HOC6H4NO2) yield a nitrophenoxy radical which is assumed to react with $O_3$ and $NO_2$ like phenoxy radical (C6H5O). We have made this point clearer in the revised manuscript.

*Table 3: Can you confirm that these are area weighted? The surface ozone seems a bit high compared to other models I've seen.*

Reply:
Yes, they are. We have added this information in the table caption. We share the impression of the reviewer that the model computes high levels of surface ozone. We are addressing the general overestimation of tropospheric ozone by, among others, improvements of the dry deposition scheme lacking the non-stomatal sink (Emmerichs et al., 2020) and the explicit modelling of the ozone sink in cloud droplets (Rosanka et al., 2020) and deliquescent aerosols.

**References**

Atkinson, R. and Arey, J.: Atmospheric degradation of volatile organic compounds, Chem. Rev., 103, 4605–4638, https://doi.org/10.1021/cr0206420, 2003.

Bates, K. H. and Jacob, D. J.: An Expanded Definition of the Odd Oxygen Family for Tropospheric Ozone Budgets: Implications for Ozone Lifetime and Stratospheric Influence, Geophysical Research Letters, 47, e2019GL084 486, https://doi.org/https://doi.org/10.1029/2019GL084486, URL https://agupubs.onlinelibrary.wiley.com/doi/abs/10.1029/2019GL084486, e2019GL084486 10.1029/2019GL084486, 2020.

Birdsall, A. W., Andreoni, J. F., and Elrod, M. J.: Investigation of the role of bicyclic peroxy radicals in the oxidation mechanism of toluene, J. Phys. Chem. A, 114, 10 655–10 663, https://doi.org/10.1021/jp105467e, 2010.

Chen, J., Wenger, J. C., and Venables, D. S.: Near-ultraviolet absorption cross sections of nitrophenols and their potential influence on tropospheric oxidation capacity, J. Phys. Chem. A, 115, 12 235–12 242, https://doi.org/10.1021/jp206929r, 2011.

Emmerichs, T., Kerkweg, A., Ouwersloot, H., Fares, S., Mammarella, I., and Taraborrelli, D.: A revised dry deposition scheme for land-atmosphere exchange of trace gases in ECHAM/MESSy v2.54, Geoscientific Model Development Discussions, 2020, 1–32, https://doi.org/10.5194/gmd-2020-139, URL https://gmd.copernicus.org/preprints/gmd-2020-139/, 2020.

Heiden, A. C., Kobel, K., Komenda, M., Koppmann, R., Shao, M., and Wildt, J.: Toluene emissions from plants, Geophysical Research Letters, 26, 1283–1286, https://doi.org/https://doi.org/10.1029/1999GL900220, URL https://agupubs.onlinelibrary.wiley.com/doi/abs/10.1029/1999GL900220, 1999.

Henze, D., Seinfeld, J., Ng, N., Kroll, J., Fu, T.-M., Jacob, D. J., and Heald, C.: Global modeling of secondary organic aerosol formation from aromatic hydrocarbons: high-vs. low-yield pathways, Atmos. Chem. Phys., 8, 2405–2420, 2008.

Jöckel, P., Kerkweg, A., Pozzer, A., Sander, R., Tost, H., Riede, H., Baumgaertner, A., Gromov, S., and Kern, B.: Development cycle 2 of the Modular Earth Submodel System (MESSy2), Geosci. Model Dev., 3, 717–752, https://doi.org/10.5194/GMD-3-717-2010, 2010.

Lin, G., Penner, J. E., Sillman, S., Taraborrelli, D., and Lelieveld, J.: Global modeling of SOA formation from dicarbonyls, epoxides, organic nitrates and peroxides, Atmos. Chem. Phys., 12, 4743–4774, https://doi.org/10.5194/acp-12-4743-2012, 2012.

Rosanka, S., Sander, R., Franco, B., Wespes, C., Wahner, A., and Taraborrelli, D.: Oxidation of low-molecular weight organic compounds in cloud droplets: global impact on tropospheric oxidants, Atmospheric Chemistry and Physics Discussions, 2020, 1–33, https://doi.org/10.5194/acp-2020-1041, URL https://acp.copernicus.org/preprints/acp-2020-1041/, 2020.

Sander, R., Jöckel, P., Kirner, O., Kunert, A. T., Landgraf, J., and Pozzer, A.: The photolysis module JVAL-14, compatible with the MESSy standard, and the JVal PreProcessor (JVPP), Geosci. Model Dev., 7, 2653–2662, https://doi.org/10.5194/GMD-7-2653-2014, 2014.

Vereecken, L.: Reaction Mechanisms for the Atmospheric Oxidation of Monocyclic Aromatic Compounds, chap. Chapter 6, pp. 377–527, World Scientific Publishing, https://doi.org/10.1142/9789813271838_0006, URL https://www.worldscientific.com/doi/abs/10.1142/9789813271838_0006, 2019.

WMO: Definition of the tropopause and of significant levels, URL https://library.wmo.int/doc_num.php?explnum_id=6960, 1957.

---

## Author Response (AR2)

**Reply to Editor's comment**

Taraborrelli *et al.*

January 8, 2021

Dear Editor,

thank you very much for your additional comments on our revised manuscript. It further helped us improving the manuscript. Please find below a point-by-point response to your comments.

> *Dear authors,*
>
> *Thank you for your thoughtful responses to reviewer comments. It appears that you have adequately dealt with the concerns raised and I will be happy to accept your paper for publication with some minor edits to help clarify a few of your additions to the text.*

Reply:
We are very glad to know about your positive judgement.

> *Specifically:*
> *1. In response to reviewer 1, in the paragraph that starts with "Figure 5" you have a sentence 2/3 of the way through the paragraph that starts with "However, the latter in the EMAC model..." I don't know what you mean by the latter since the preceding sentence doesn't give two kinds of anything. I imagine you are talking about either OH or the methane lifetime but I would just say that explicitly rather than "the latter".*

Reply:
We meant methane lifetime and we now mention it explicitly.

> *2. In response to the comment about grid resolution, I don't think I follow your logic for why higher resolution might reverse the effects of aromatics on ozone. I understand the previously published effects for NOx because ozone production is very non-linear in NOx so an intense plume behaves quite differently than a dilute one. But ozone production at a given NOx level will always increase with increasing VOC so I don't really understand how grid resolution could invert the response you show. For example, in the higher resolution case, for coemitted NOx and aromatics, then NOx-HOx interactions will, as you say, generate more HNO3 and less O3 than in the lower resolution case. But the aromatics will simply survive longer since near-field OH is being consumed by reaction with NOx so then they will disperse farther and the effect of aromatics might actually be more similar between the two resolutions than is the case for NOx. I may not be expressing my confusion clearly but I think it would be better if you could be more explicit in the chain of effects that lead you to expect higher resolution to lead to lower ozone enhancements resulting from a given aromatic emission.*

Reply:
We are grateful to the editor for having critically questioned what we have written about the expected impact of model spatial resolution on the results. We also apologize for the confusion. We had in mind the net change in ozone production ($dO_x/dt$) and not $O_3$ at the surface. The motivation is

a preliminary result we got by running the model at two different spatial resolutions. For a boreal summer going from 300km (Figure 1) to 60 km (Figure 2), $dO_x/dt$ noticeably becomes:

- more negative in some urban regions in the northern hemisphere like central Europe, eastern US, south Korea and Japan

- very small in some areas in the tropics like south of southern America and south of the Himalaya mountains.

[Figure]

Figure 1: Net change of ground-level $O_x$ (mol/mol/s) at 300 km resolution.

[Figure]

Figure 2: Net change of ground-level $O_x$ (mol/mol/s) at 60 km resolution.

This effect is outside the scope of the present manuscript and would deserve a detailed investigation of the $O_x$ and $HO_x$ budgets from very expensive model simulations at high resolution. We obviously agree with the editor about the increase in ozone production with increasing VOC except at very low $NO_x$. Therefore, we have removed from the text the part claiming a possible reversal of the

surface ozone change. Nevertheless, we think that the reduction in tropospheric ozone when using a 2-way nesting for North America, Europe and East Asia like Yan et al. (2016) is consistent with the established understanding of tropospheric ozone chemistry. For instance, such a 2-way nesting would make the predicted increase of surface ozone by aromatics smaller and consequently results in a more negative change in tropospheric ozone compared to what our study is suggesting. Thus, we have adjusted the relevant text from

Therefore, at much higher spatial resolutions we expect that the enhancement of surface ozone by aromatics in those regions (Fig. 6) to be greatly reduced if not reverted.

to

Therefore, at much higher spatial resolutions the predicted enhancement of surface ozone by aromatics in those regions (Fig. 6) might be reduced.

*3. You also have several type-o's in your next response: "There in fact consistent" should be "There IS in fact consistent", you're missing some spaces in "HO2 and RO2" and "This uncertainty limit" should be "This uncertainty limitS". Similar things are going on in subsequent responses, please just be sure to carefully proofread any new text for format, grammar and clarity.*

Reply:
We apologize for these type-o's. We have corrected them. We also made a few other minor changes. **(to do)**

**References**

Sillman, S., Logan, J. A., and Wofsy, S. C.: The sensitivity of ozone to nitrogen oxides and hydrocarbons in regional ozone episodes, J. Geophys. Res., 95D, 1837–1851, https://doi.org/10.1029/JD095iD02p01837, 1990.

Yan, Y., Lin, J., Chen, J., and Hu, L.: Improved simulation of tropospheric ozone by a global-multi-regional two-way coupling model system, Atmospheric Chemistry and Physics, 16, 2381–2400, https://doi.org/10.5194/acp-16-2381-2016, URL https://acp.copernicus.org/articles/16/2381/2016/, 2016.

**Reply to RC1 on acp-2020-461**

Taraborrelli *et al.*

January 6, 2021

Dear Anonymous Referee #1,

thank you for your thoughtful review of our manuscript. It helped us improving the manuscript considerably. Please find below a point-by-point response to your comments.

> **Description**
> *This manuscript describes the changes in trace gas concentrations that occur when emissions of monocyclic aromatic compounds are included in a global general circulation / atmospheric chemistry model. Many recent studies have pointed out the strong influence of aromatic compounds on local and regional air quality, especially in East Asia, which makes this manuscript a timely assessment of their treatment in models and their global impacts. Here, the authors incorporate the MECCA mechanism into EMAC and diagnose the implications of aromatic chemistry by comparing a simulation with aromatic emissions turned on to one with aromatic emissions turned off. Unsurprisingly, they find that aromatic compounds contribute substantially to the global budgets of glyoxal and methylglyoxal – two of their main oxidation products and important SOA precursors – while their contributions to the budgets of smaller and more ubiquitous carbon-containing compounds, such as HCHO and CO, are smaller relative to the background. For budgets of HOx, NOx, and ozone, the effects are more complex; while aromatic emissions have small global effects (which, intriguingly, sometimes contradict those found by previous studies), their local effects can be moderately substantial and can vary in sign depending on local chemistry.*

Reply:
We appreciate Referee #1 for the accurate and synthetic summary of our manuscript.

> **General comments**
> *The manuscript is straightforwardly written, well laid-out, and presents its findings clearly. Furthermore, because the authors use a highly comprehensive mechanism based on MCM, the chemical outcomes of aromatic oxidation are likely more robust than previous studies. However, it appears from looking more closely at the mechanism that some of the larger aromatic species (e.g. xylene and ethylbenzene) are not actually treated explicitly, but are oxidized in pathways identical to those of toluene with a "LCARBON" counter species denoting the carbon imbalance that arises from this treatment. This should be more explicitly described in the manuscript, which appears to imply that all aromatic species are treated independently by providing their individual emissions.*

Reply:
Indeed we have not made this aspect clear. In the model description (Section 2) we now stress this approximation and the error it may result from for prediction of stable products and low volatile compounds contributing to SOA formation.
We have added the text below at L75:

In short, the MCM schemes for benzene and toluene were taken. Following the approach of Taraborelli et al. (2009), short-lived intermediates were replaced with their stable products and isomeric peroxy radicals were lumped preserving the yield of stable products. Initial oxidation steps of aromatics other than benzene and toluene are considered and products replaced by the analogous toluene oxidation products. This approximation inherently introduces an error with respect to the formation of larger and low volatile products. The carbon mass that is not accounted for with this approximation is however tracked by introducing the counter LCARBON for the difference of carbon atoms between the oxidation products of larger aromatics and toluene.

> *Another general concern is that in many places model outcomes are simply described without sufficient discussion of their causal pathways, which strongly diminishes the usefulness of these findings. Further, many of the important effects of these model outcomes are only touched on without any quantitative discussion – e.g., the changes to methane lifetime and SOA formation from glyoxal and methylglyoxal.*

Reply:

We have put significant efforts in discussing the results but we are happy to improve the manuscript in this respect by following specific indications. Concerning the first specific effect that Referee #1 pointed to, we acknowledge that the quantification of the impact on methane lifetime could have been given more space than a short mention without referring to Table 4. Therefore, we have extended the paragraph L129-132 by pointing explicitly to Table 4 and shortly discussing the changes in OH and CH4 lifetime in the two hemispheres.

The paragraph

Figure 5 shows the annual zonal mean changes of the OH mixing ratio. The changes are most pronounced in the NH upper troposphere where reductions range from $7\%$ to $20\%$. This helps bringing the model-simulated inter-hemispheric OH asymmetry closer to that derived from observations (Lelieveld et al., 2016). Globally, aromatics oxidation reduces OH by $7.7\%$ and consequently increases methane lifetime.

has been extended:

Figure 5 shows the annual zonal mean changes of the OH mixing ratio. The changes are most pronounced in the NH upper troposphere where reductions range from $7\%$ to $20\%$. These predicted changes are associated to similar reductions in $NO_x$. In fact, the upper troposphere is in general $NO_x$-limited and the oxidation of aromatics enhances the formation $N_2O_5$ and $HNO_3$ which are lost heterogeneously. This leads to an effective removal of $NO_x$ from the gas phase and lowers the radical production. The change in hemispheric burdens of OH are consistent with this picture (Table 4). This moderately helps bringing the model-simulated inter-hemispheric OH asymmetry closer to that derived from observations (Lelieveld et al., 2016). Globally, aromatics oxidation reduces OH by $7.7\%$ and consequently increases methane lifetime by about $5.5\%$. The changes are more pronounced in the northern hemisphere where aromatics are mostly emitted (Table 4). However, in the EMAC model methane lifetime remains significantly lower than the ACCMIP multi-model mean and the observational-based estimates (Naik et al., 2013). Coarse model spatial resolutions (about 200 km) are known to result in an overestimation (underestimation) of global mean OH (methane lifetime) of at least $5\%$ (Yan et al., 2016). This is due to a less efficient conversion of $NO_x$ to $NO_y$ when strong pollutant emissions are artificially diluted in the model grid boxes. This aspect certainly has a larger impact on the inter-hemispheric OH asymmetry in atmospheric models that is in contrast to observational estimates (Patra et al., 2014).

With respect to the second specific effect Referee #1 pointed to, SOA formation from $\alpha$-dicarbonyls, we think this is beyond the scope of the present manuscript for the following reason. The simulations were performed with a VBS-based approach to model condensation of organic vapours, at the time

of writing of the manuscript no representation of oligomer formation from (methyl)glyoxal was implemented in the EMAC model yet. This is now implemented explicitly for cloud droplets (Rosanka et al., 2020) and its effect is planned to be assessed in a subsequent study together with the contribution of reactive uptake of epoxides from isoprene and aromatics. We have extended paragraph L201-209 by adding this explanation.

Specifically, we have extended

These changes are of significance for the model SOA budget since these two dicarbonyls are estimated to produce a large fraction of SOA by cloud processing (Lin et al., 2012).

These changes are of significance for the model SOA budget since these two dicarbonyls are estimated to produce a large fraction of SOA by cloud processing yielding low-volatile oligomers (Lin et al., 2012). However, a model assessment of SOA formation from $\alpha$-dicarbonyls is beyond the scope of this study. The reason is that, although the simulations were performed with a VBS-based approach to model condensation of organic vapours, the EMAC model version used in this study has no representation of oligomer formation from (methyl)glyoxal. This has been recently implemented explicitly for cloud droplets (Rosanka et al., 2020) and its effect is planned to be assessed in a subsequent study together with the contribution of reactive uptake of epoxides from isoprene and aromatics.

> *Specific instances are pointed out in the comments below. Finally, while the short discussion of uncertainties in Section 4 is a particularly useful addition to this manuscript, it does not go far enough to provide the reader with either quantitative or qualitative bounds on the model outcomes described herein. Of particular note, more attention should be paid to:*
> *(a) the effects of grid resolution – given that there's a lot of spatial heterogeneity in model outcomes, and the effects even change sign depending on local conditions, is 1.875 square enough to resolve this chemistry? What outcomes might be masked by the artificial mixing that occurs in such large-scale grid boxes?*

Reply:
We thank the reviewer for pointing to this important aspect that, indeed, we have not mentioned in our manuscript. We gladly take this chance to discuss the influence of (horizontal) spatial resolution on the predicted changes of trace gas levels. We now shortly mention it in our extension of paragraph L129-132 concerning OH and methane lifetime . Moreover, we add a whole paragraph in Section 4 on model uncertainties in which we frame and formulate our expectation for modelled $HO_x$, $NO_x$ and $O_3$.

The spatial resolution of atmospheric models has a significant influence on the predicted levels of oxidants and nitrogen oxides. Generally in polluted regions the coarser the resolution the larger the ozone production per molecule of $NO_x$ will be (Sillman et al., 1990). This is due to the artificial dilution of strong NOx emissions which, in reality, is efficiently converted to $NO_y$ by reacting with $HO_x$. For instance, reducing the spatial resolution over the polluted North America, Europe and East Asia with a two-way nested regional model led to a 9.5 % reduction in the global tropospheric ozone burden (Yan et al., 2016). We have shown that at our model resolution of $1.875° \times 1.875°$ aromatics are estimated to induce important increases in $HO_x$ (Fig. 3) and decreases in $NO_x$ (Fig. 10 and 11) over continental polluted regions. Therefore, at much higher spatial resolutions the predicted enhancement of surface ozone by aromatics in those regions (Fig. 6) might be reduced. Based on the results by Yan et al. (2016) we expect this effect to translate in a significant enhancement of the tropospheric ozone reduction reported in this study (Sect. 3.2). A quantification of the model resolution effect on chemical regimes is at the moment computationally prohibitive with our very large chemical scheme running in the global EMAC model.

> *(b) uncertainties in chemical mechanisms, especially the newly implemented ones de-*

*scribed in the bullet point list in Section 2 –while there is some discussion in Section 4 of the uncertainty in the phenoxy + O3 reaction, it should be put into a larger context and more explicitly tied to the uncertainty bounds of model outcomes that might be expected given the uncertainties in the chemical mechanism;*

Reply:
We agree with Referee #1. We have expanded Section 4 by discussing the uncertainties associated with the limitations of currently accepted oxidation mechanisms like the MCM. Specifically, we now mention the uncertainty on the epoxide formation pathway that is treated as direct in the MCM. This likely involves intermediate steps implying an epoxide yield dependent on $NO_x$ and $HO_x$ levels (Vereecken, 2019).

We have added the paragraph below:

Another source of uncertainty is the direct formation of epoxide upon addition of OH and subsequently by $O_2$ as implemented in the MCM ranging from 11.8 %, for benzene, to 24 %, for trimethylbenzene (Bloss et al., 2005b,a). There is in fact consistent theoretical evidence that the epoxide formation pathway passes through a second $O_2$-addition. This implies that the epoxide yield likely depends on the abundance of NO, $HO_2$ and $RO_2$ (Vereecken, 2019, and references therein). This uncertainty limits the reliability of the predicted SOA formation from reactive uptake of epoxides by aerosols (Paulot et al., 2009).

*and lastly (c) uncertainties in emissions – while these are also discussed briefly in Section 4, the scope of the discussion is very limited and does not leave readers with any quantitative understanding of how well the emission totals are known, what their overall inter-annual variability might be, and how this could affect the model outcomes described in Section 3. I am not trying to argue that the authors need to perform additional sensitivity simulations, but the uncertainties merit a more lengthy,detailed, and quantitative description than is provided here.*

Reply:
We agree with Referee #1 and therefore we expanded paragraph L243-247 of Section 4 by discussing magnitude and inter-annual variability of aromatics emissions from biomass burning, anthropogenic activities and terrestrial vegetation.

We have added the paragraph below:

However, it appears that the two major contributions to this variability are the peat fires in Indonesia and boreal forest fires, which are strongly favoured by El Nino and heat waves, respectively. An early estimate of anthropogenic emissions of aromatics gave 16 TgC/a, (Fu et al., 2008). Two relatively recent datasets yield about 50% higher emissions being 23 TgC/a for RCP (Cabrera-Perez et al., 2016) and 22 TgC/a for EDGAR 4.3.2 (Huang et al., 2017). The latter is used in this study and lacks the biofuel burning emissions of phenol, benzaldehyde and styrene. Inter-annual variability of anthropogenic emissions of aromatics is is not well known but the decadal trends are known to be negative since the 1980s (Lamarque et al., 2010). Aromatics emissions from terrestrial vegetation have been long neglected or considered very low. However, Misztal et al. (2015) suggested that aromatics emissions from biogenic sources may rival those from anthropogenic ones. In this study we used the same emission algorithm used in Misztal et al. (2015) but get much lower emissions for toluene (about 0.3 vs. 1.5 TgC/a). However, Misztal et al. (2015) suggest that emissions of aromatics and benzenoid compounds may be in the 1.4-15 TgC/a range. The major contributors are toluene and some benzenoids (oxygenated aromatics). The latter are mainly emitted during blossoming and stress-induced reactions by plants. The variability of their emissions is not very well quantified. For instance, the MEGAN model calculates their emission strengths based of the ones for carbon monoxide (Tarr et al., 1995).

***Specific Comments***
*Additional (more specific)questions about uncertainty are given below.*
*L 19 - Does the 200-300 Tg/yr refer to Tg O3?*

Reply:
Yes, it does. We believe that in the relative sentence it is clear that we refer to $O_3$.

*L 69-72 - Because the primary findings of this paper rely so heavily on the magnitude of these emissions, some small discussion of their uncertainty is warranted. Do the sources from which the inventories were derived describe the range of plausible emission amounts? Do different anthropogenic or biomass burning inventories give different emission amounts? EDIT - I see this is partially addressed in Section 4 (though it would be useful here to direct the reader to the later discussion of uncertainties). However,the short paragraph about emissions uncertainties still lacks quantitative detail, and most of the questions above still remain unanswered. Also, can a numerical range of inter-annual variability of total pyrogenic aromatic emissions be provided?*

Reply:
Referee #2 made a similar point in the general comment above. We agree and have expanded the relative paragraph in Section 4 and mentioned in our reply above.

*L 75 - It would be helpful here to give detail not only on the additions that have been to the mechanism, but also on generally what simplifications were made to MCM to arrive at MECCA. I realize that's available in Cabrera-Perez et al. 2016, but the reader should be able to get a sense of the methodology here without having to fully read another paper. In particular, it is problematic that this manuscript implies a full detailed mechanism for the suite of aromatics shown in Table 1 when in fact many of them just use the same oxidation schemes.*

Reply:
We thank Referee #1 for making us aware of this aspect. For that we have added a short description of the simplifications made to the MCM mechanism for benzene and toluene. This is now combined with the mentioning of the LCARBON species counter as requested in your general comment.

*L 122-124 - This is not a complete explanation of the OH increase in high-NOx regions. It is stated that the increased OH is "caused by the reaction of NO with HO2", but it was also stated two lines previously that NO decreases in these regions. Does HO2 increase enough as to offset both the NO decrease and the OH decrease through direct reaction with aromatics? What causes such a pronounced HO2 increase? Further,Figures 4 and 5 are not particularly useful to the reader without an explanation of why these effects occur. What causes the seasonal variability in the effects of aromatics on OH? Why are they strongest in the upper troposphere?*

Reply:
The increase in $HO_2$ indeed overcompensates for the decrease in NO resulting in enhanced OH levels over regions where radical production is not $NO_x$-limited. The $HO_2$ production from VOC oxidation is a well established knowledge in atmospheric chemistry. In *AROM* compared to *onlyMCM* this production is further enhanced by the photolysis of ortho-nitrophenols and benzaldehyde that we have mentioned in the manuscript. We have modified the manuscript in order to make this point clear.

The paragraph

We find a positive correlation between OH and anthropogenic emissions in these regions but a negative correlation in the low-$NO_x$ CAF region. The increased OH in the high-$NO_x$ regions is mainly caused by the reaction of NO with $HO_2$.

has been modified to

We find that inclusion of aromatics emissions leads to an increase OH in these regions but to decrease in the low-$NO_x$ CAF region. The increased OH in the high-$NO_x$ regions is mainly caused by the reaction of NO with $HO_2$. The production of OH from this important reaction is enhanced by the significant $HO_2$ formation in aromatics oxidation. Compared to *onlyMCM* the *AROM* simulation has additional $HO_2$ production from the photolysis of ortho-nitrophenols (R1) and benzaldehyde (Sect. 2). The enhanced $HO_2$ levels (not shown) overcompensates the negative changes in NO (see Sect. 3.2).

With respect to Figure 4 we have added a brief explanation of why the largest decrease in planetary boundary OH is computed for the NH.

In general enhancements are predicted for regions where radical production is not $NO_x$-limited. In the NH there obviously more such regions compared to the SH. However, the largest decrease in the planetary boundary OH is computed for the NH where most of the emissions of aromatics are located.

With respect to Figure 5 we have added an explanation for the reduced OH levels in the upper troposphere. The predicted changes are associated to similar reductions in $NO_x$. In fact, the upper troposphere is in general $NO_x$-limited and the oxidation of aromatics enhances the formation $N_2O_5$ and $HNO_3$ which are lost heterogeneously. This leads to an effective removal of $NO_x$ from the gas phase and lowers the radical production.

We have extended the text below

Globally, aromatics oxidation reduces OH by $7.7\%$ and consequently increases methane lifetime.

with

Globally, aromatics oxidation reduces OH by $7.7\%$ and consequently increases methane lifetime by about $5.5\%$. The changes are more pronounced in the northern hemisphere where aromatics are mostly emitted (Table **??**). However, the latter in the EMAC model remains significantly lower than the ACCMIP multi-model mean and the observational-based estimates (Naik et al., 2013). Coarse model spatial resolutions (about 200 km) are known to result in an overestimation (underestimation) of global mean OH (methane lifetime) of at least $5\%$ (Yan et al., 2016). This is due to a less efficient conversion of $NO_x$ to $NO_y$ when strong pollutant emissions are artificially diluted in the model grid boxes. This aspect certainly has a larger impact on the inter-hemispheric OH asymmetry in atmospheric models that is in contrast to observational estimates (Patra et al., 2014).

> *L 123 - Is this "positive correlation" a quantitative effect, diagnosed by some sort of regression analysis (across time? or just regions?), and if so, can it be explained in more detail here? If this "positive correlation" is just meant to say that including anthropogenic emissions causes an increase in OH, then this phrasing is misleading – better just to state simply that OH is higher in AROM than NOAROM in these regions. Also,are these correlations really diagnosed from anthropogenic emissions in particular, or from the inclusion of all aromatic emissions (including biogenic and pyrogenic)?*

Reply:
We apologize for this misleading formulation. We have modified the sentence as suggested by the Referee by removing the use of the word "correlation".

> *L 132 - The increased methane lifetime is likely to be of great interest to readers. Can it be quantified?*

Reply:
It is quantified and also broken down for the two hemispheres in Table 4. As mentioned in the reply to the general comments above, we expanded paragraph L129-132 in which we have made an explicit reference to Table 4 and quantify the change in calculated methane lifetime.

*L 138-140 - The same comment above (L 123) applies here to the "correlation" phrasing.*

Reply:
We apologize again for the misleading formulation and have changed the text similarly as mentioned in the answer above.

*L 141 - It appears Figure 9 is discussed here in the text before Figures 7 (L 143) and 8 (L157)*

Reply:
Thank you for pointing to this. We fixed it in the revised manuscript.

*L 141-145 - The same comment above (L 122-124) applies here; why do these seasonal and zonal patterns arise?*

Reply:
The inter-hemispheric changes in $O_3$ are indeed similar, although much less pronounced, and not independent from the changes in OH. In the revised manuscript we describe it and stress more the connection to changes in OH.

We have added the sentence

Like for the OH levels, the inter-hemispheric asymmetry in the emission of aromatics determines the higher$O_3$ decrease in the NH compared to the SH.

*L 153 - Can some more quantitative description of the variation with tropopause definition (and a description of the definition itself) be included here? What specifically is meant by "robust"? Figure 7 makes it look instead like there are large absolute differences right at the tropopause, which would imply that the definition might be highly important.*

Reply:
We have extended the caption of Table 4 by listing all 6 different definitions and diagnostic "tropopauses" that are calculated by EMAC with the MESSy submodel TROPOP (Jöckel et al., 2010). In the caption we also report that the results do not change by more than 0.05 %.

The caption of Table 4 has been extended with the text below

Tropospheric burdens were reckoned using six different tropopause definitions (provided by the TROPOP submodel, see Jöckel et al. (2010) for details): 1,2 surfaces of $O_3$ mixing ratio of 125 and 150 nmol/mol, respectively, 3) WMO definition (WMO (1957)), 4) dynamic PV-based (3.5 PVU potential vorticity surface, sought within 50–800 hPa), 5) climatological (invariable zonal profile, i.e. 300-215•(cos(latitude))$^2$ hPa) and 6) the combined definition (WMO tropopause within 30°N–30°S, otherwise dynamic PV-based tropopause). The latter definition is used by default in EMAC and in this work. Estimated changes to tropospheric $O_3$ burden are identical within 0.05 % between the available definitions.

*L 154 - What is meant by "these changes", and why does the explanation provided here differ from those above (L 134-136)?*

Reply:
We acknowledge the inconsistency and the lack of clarity of this formulation. What we wanted to highlight is the direct ozone loss by reaction with (substituted) phenoxy radicals that we find to play as an **additional** and previously overlooked ozone sink at global scale. We have now reformulated the corresponding sentence in this sense.

We have changed the sentence

These changes are associated with the enhanced direct ozone loss by reactions with organic compounds.

to

The changes in ozone are caused by perturbations of the radical production in different $NO_x$ regimes but also by the direct ozone loss in reactions with organic compounds.

*L 159 - Extra close-parentheses.*

Reply:
Corrected.

*L 143-164 - This paragraph is long and covers a wide range of different topics; consider splitting it up? Also, the sentences around L 156-160 seem to be in an odd order; the sentence beginning "However, with aromatics" seems like it should be followed by the sentence beginning "Ozone is known to react", while the two intervening sentences seem like a non sequitur.*

Reply:
We thank the Referee #1 for this comment. We now split the long paragraph where we start discussing the direct losses of ozone in the oxidation of organic compounds. Our sentence beginning with "However, with aromatics" is meant to stress that all the other global atmospheric chemistry models do not represent additional direct loss of ozone in VOC oxidation. We hope that our explanation clarifies the issue.

*L 161-164 - How well is this chemistry known? The cited study describes the reactions of the phenoxy radical with O3 and with other phenoxy radicals, but presumably this is also in competition with many other reactions, including possible unimolecular re-arrangements or decomposition. Have competitive studies been reported? If not, can some estimate of competing reactions rates be used along with the uncertainty bounds of the phenoxy + ozone reaction rate to determine some level of certainty for this discussion here? Considering how strong the simulated effect is, and how its catalytic nature under some conditions could magnify even small errors, some discussion of uncertainty is warranted. EDIT - I see this is partially addressed below in section 4 (it'd be nice to have some parenthetical here directing the reader to the later discussion on uncertainties). Some questions remain, though – most notably why the rate constant"has to be regarded as a lower limit" (L 227) and whether competing reactions might also be uncertain.*

Reply:
We thank the Referee #1 for this useful comment. The chemistry of phenoxy radicals is indeed not

very well known. The rate constant for the reaction with ozone has been determined at only ambient temperature. The latter is a lower limit because of the nature of the kinetic experiments and analysis conducted by Tao and Li (1999). Phenoxy radical is very stable radical and the only other known sink is the reaction with $NO_2$, which yields ortho-nitrophenols. The rate constant of the latter reaction is about one order of magnitude higher. However, ozone is very often more abundant than $NO_2$ by more than an order of magnitude. This makes ozone to our knowledge the major atmospheric sink for phenoxy radicals. The reaction with NO is reversible and not considered neither in MCM nor in our mechanism. We have mentioned these additional sinks for phenoxy in the manuscript. However, we did not mention the relative magnitude of the rate constants and neither did we make a statement about ozone being the major atmospheric sink of phenoxy radicals. We now mention this aspect in the revised manuscript and have added a reference to the discussion of the mechanistic uncertainties in Section 4.

We have modified the sentence in L166-167

This ozone loss is enhanced by phenoxy radical production in the R2 reaction and the subsequent loss of odd oxygen by $NO_3$ photolysis and $N_2O_5$ heterogeneous loss

to

Although the known rate constant for reaction R3 is about one order of magnitude lower than the others, the high abundance in the atmosphere makes ozone the major sink of (substituted) phenoxy radicals. This direct ozone loss in reaction R3 is enhanced by phenoxy radical production in reaction R2 and the concurrent loss of odd oxygen by $NO_3$ photolysis and $N_2O_5$ heterogeneous loss

*L 164 - No period (assuming this is the end of the sentence).*

Reply:
Yes, we now close the sentence with a column.

*L 173-174 - How does nitrophenol photolysis come into play here for the phenoxy radicals? R1 implies either that nitrophenol photolysis does not form phenoxy radicals or that the mechanism skips the phenoxy radical step and goes directly to decomposition products.*

Reply:
The HONO-channel in the photolysis of ortho-nitrophenols is predicted to form phenyloxy radicals which likely rearrange to a 7-membered ring radical and further decomposes (Vereecken et al., 2016). Formation of phenoxy radicals from photolysis of ortho-nitrophenols is not skipped. However, we acknowledge that the OH-channel, which may be much more important of the HONO-channel, produce nitrosophenoxy radicals might efficiently react with ozone similarly as phenoxy radicals. However, this chemistry is unknown. In Section 3.2 we now refer to Section 4 where we already discuss these mechanistic uncertainties.

*L 186-187 - This phrasing does not make it particularly clear which simulation has the higher NO3 concentrations.*

Reply:
Indeed it is not clear. We have modified the first part of the sentence by starting with "Relative to NOAROM, in AROM ..."

*L 188 - Why does this effect occur in places with pyrogenic aromatic emissions, while the HOx and O3 effects described previously are more strongly associated with anthropogenic emissions? The same question applies to the HONO increase on L 191.*

Reply:
We are not sure why these decreases are predicted and suppose is a result of a complex interplay of multiple factors. Thus, we refrain to make statements not backed by a solid understanding. We added to the manuscript that these changes are modest and stress more the widespread increase of $NO_3$ levels by the reaction of phenyl peroxy radicals with $NO_2$.

We have modified the sentence at L189 from

Comparing *AROM* to *NOAROM*, the global average of the nighttime species $NO_3$ increases by more than 7 % (Tab. 3).

to

However the latter seems to dominate and cause a significant and widespread increase in the predicted $NO_3$ levels. Relative to *NOAROM*, in *AROM* the global average of the nighttime species $NO_3$ increases by more than 7 % (Table 3).

> L 198-200 - What is the explanation for these findings?

Reply:
We think this is due to the concurrent enhancement of OH levels which "curb" the enhancement of HCHO in China, Europe and US. We have modified the text in order to express this explanation.

We have expanded the relative sentence to

There are, however, regional differences that are moderate because of the concurrent enhancement of the HCHO sink by reaction with OH.

> L 208-209 - Was the effect on the SOA budget quantified here? It seems this effect would be of great interest to some readers.

Reply:
No, it was not quantified in this study. However, it will be subject of future studies in which the production of oligomers from dicarbonyls in the condensed phase is represented. Recently, we have added the cloud processing of dicarbonyls to the scavenging module of MESSy (Rosanka et al., 2020) and will evaluate the impact of it on the SOA budget in a future study.

> L 218 - It seems odd that so much detail is shown in the many other figures for other species discussed in this manuscript, but this interesting finding for CO is not shown. Can a CO figure be added, and can an explanation be given for these spatial effects?

Reply:
We understand Referee #1 but we had the feeling we had already too many figures in the manuscript. We are happy to add to the manuscript the figure showing the zonal mean differences for CO which peak in the NH UTLS.

> L 230-231 - It was earlier implied (L 173-174) that the model *does* account for the effects of photolysis of nitrophenols. Can this be clarified?

Reply:
The model accounts for the known photolysis of ortho-nitrophenols yielding HONO. However, our

model does not account for the less known OH-channel potentially yielding nitrosophenoxy radicals (Vereecken et al., 2016), which might react similarly as phenoxy radicals. At the time of finalizing the chemical mechanism we were not aware of the results by Vereecken et al. (2016). Nevertheless, we state clearly in Section 4 that our model lacks photolysis of nitrophenols yielding phenoxy radicals. In this section we have replaced "reforming phenoxy radicals" with "forming nitrosophenoxy radicals".

> ***Technical comments***
> *Table 5 is cut off by the end of the page.*

Reply:
We apologize for this inconvenient. The table is not cut off in the ACP article layout. We report the complete table here at the end of the document (see Table 1).

> *Can the OH mixing ratios in Figures 3-5 be expressed instead in the more commonly used concentration units?*

Reply:
We know that OH abundance in the atmosphere is usually expressed in molec cm$^{-3}$. However, doing it in our manuscript would introduce an exception and an inconsistency to the way we present the results. For this reason we would like not to modify the units.

> *Figure 4 is very confusing. What are the units on the left plot? Why are there two very close blue and red lines in the left plot? Why does the caption reference solid and dashed lines in the right plot when only solid lines exist? The same questions apply to Figure 9.*

Reply:
In all Figures of this manuscript, the plotted values correspond to the original output values in mol/mol multiplied by ten to a certain power. For example, the values on the vertical axis of Fig. 4 (left) stand for mol/mol multiplied by 1e+14. Since this notation is not intuitive, we changed the titles of the plots and added the unit (e.g. "x $10^{-11}$ mol/mol").
The caption is indeed not clear, as the dashed and solid lines are only in the left plot of Fig. 4 and Fig. 9 (they distinguish the results of AROM and NOAROM). We changed the caption of Fig. 4 to: "
[revised manuscript text omitted]

**Reply to RC2 on acp-2020-461**

Taraborrelli *et al.*

December 23, 2020

Dear Alexander Archibald (Referee #2),

thank you for taking the time to review of our manuscript. It gave us the chance to improve the manuscript significantly.
Please find below a point-by-point response to your comments.

> ### Description
> *Taraborrelli et al. provide an updated mechanism for the oxidation of aromatics in the EMAC model and a summary of the impacts of this update on key trace gases in the troposphere. This is a generally well written and well executed study but I have several minor comments in the attached pdf and a few more major comments before recommending publication.*

> ### Major comments
> *Comparison to observations is lacking which limits the sense I get that the changes are in anyway in the right direction. For example, the changes in surface ozone and NO2 in EAS are large and I would imagine significant. It would be good to see how these compare with observations. Whilst I agree with the previous reviewers comments about model resolution and whilst there may well be structural errors in EMAC that mean that even with a better representation of the chemistry the comparison to observations is worse, I feel some comparison to observations is warranted.*

Reply:
We agree with the referee that comparison with observations would strengthen and focus the manuscript. However, a comprehensive evaluation of the EMAC model with the complex organic chemistry (MOM) and against satellite retrievals of $O_3$ and $NO_2$ is in preparation and will be soon submitted for peer review. A first comparison of the model results with IASI-FORLI retrievals for ozone along with a detailed $O_x$ budget is currently presented in Rosanka et al. (2020). The model results for tropospheric ozone, with the modified MCM chemistry of aromatics we present here, are clearly still too high with overestimates of up to 10 DU. This positive bias will be addressed by further improving existent parametrizations in EMAC like the dry deposition scheme (Emmerichs et al., 2020) and extending the representation of multiphase chemistry, which started with Rosanka et al. (2020), to deliquescent aerosols.

> *I also feel this will help focus the paper as currently it reads as one in which aromatics vs no-aromatics is the focus, but given we know aromatics are important (and abundant in urban environments) shouldn't the focus be Arom vs OnlyMCM? At least I find this comparison more interesting than Arom vs NoArom. It would be good if there were some figures (perhaps in an appendix) which compare the OnlyMCM and Arom scheme under idealised (Box model) conditions. Ideally this would be against laboratory data but I think even against some general scenarios it would be very useful to see how the differences implemented affect the results and then some sensitivity analysis could be performed I think quite straightforwardly to look at the impacts of some of these uncertain thermal-kinetic and photolytic processes.*

Reply:

We thank the referee for this comment. Currently, we are not setup to compare box model simulations to lab data from chamber experiments. Our *modus operandi* is to obtain an intermediate and faithful reduction of a chemical mechanism like MCM that has been widely used and tested against lab measurements. We also believe that the differences in the results between AROM and onlyMCM are interesting to show. Instead of showing box model simulation results under idealized conditions, we think that showing the global distribution of the differences (spanning many possible scenarios) between AROM and onlyMCM simulations is a synthetic and useful way to visualize the deviations across a comprehensive set of chemical regimes. We therefore added an appendix to the manuscript to enhance the discussion of the differences between the MCM mechanism and our mechanism for the aromatics. In this appendix we briefly present the differences for the main oxidants OH, $O_3$ and $NO_3$.

The text of the appendix is reported below

In this appendix the impact of the modifications to the MCM chemistry (listed in Sect. 2) on the model results are shown for the main atmospheric oxidants.

**Hydroxyl radical (OH)**

The differences at the surface are shown in Figure A1. Much of the increase in Figure 3 can be ascribed to the enhanced $HO_x$ production by photolysis of benzaldehyde (Roth et al., 2010) and HONO from R1. The latter from benzene chemistry explains the significant enhancement across the UT/LS (see Fig. A2).

**Ozone ($O_3$)**

The differences at the surface are shown in Figure A3. It can be seen that lareg part of the enhancement in surface ozone mixing ratio in Figure 6 is due to enhanced $HO_x$ production in regions that are not $NO_x$-limited. The zonal mean change in ozone is minimal and slightly positive at the tropical UT/LS (Fig. A5).

**Nitrate radical ($NO_3$)**

The differences at the surface are shown in Figure A5. It can be seen that the widespread enhancement of in Figure 12 is largely to be ascribed to the effect of phenylperoxy reaction with $NO_2$ (R2).

**Specific Comments**

*L13: Changes of what?*

Reply:
We added "of trace gas levels" to the text.

*L19: Is this a net loss? If not, is it a very important finding?*

Reply:
This is a direct loss of ozone. As it is localized in the upper troposphere where benzene is transported and NO levels are generally low, this turns into a net loss of ozone. However, model setup used in this study did not have a comprehensive set of passive tracers that allows the classical tropospheric $O_x$ budget to be computed. Nevertheless, we think that this result, the direct loss of ozone, is worth noting especially because it is clearly missing in all other global models with which the global impact of aromatics on ozone has been estimated to be positive, contrary to our study.

*L31: This is too vague. There are specific definitions of aromaticity with implications for the chemistry of compounds in this class.*

Reply:
We agree with the referee that we need to be more specific in this respect. Thus, we have changed the first sentence of the paragraph by stating that aromatics are unsaturated planar cyclic organic compounds with enhanced stability due to a strong electron delocalization.

We have modified the text

Aromatics are a subset of unsaturated organic compounds of which several are present in the atmosphere, e.g., benzene, toluene, ethylbenzene, xylenes, styrene and trimethylbenzenes.

to

Aromatics are unsaturated planar cyclic organic compounds with enhanced stability due to a strong electron delocalization. Several of them are present in the atmosphere, e.g., benzene, toluene, ethylbenzene, xylenes, styrene and trimethylbenzenes.

*L38: Add a reference for the toluene biogenic emissions.*

Reply:
We added the reference to the first reported biogenic emission of toluene by Heiden et al. (1999).

*L42: Is that true of all aromatics? i.e. benzene?*

Reply:
We agree with the referee that it might sound odd to put benzene in the category of organic compounds that have a high reactivity. We have changed the relative sentence by removing the reference to the high reactivity and expressed in more neutral terms with a range of tropospheric lifetimes.

Accordingly, we have modified the sentence

Due to their high reactivities, aromatics have relatively short atmospheric lifetimes ranging from hours to a few days.

with

Aromatics have relatively atmospheric lifetimes ranging from a few hours, e.g. for trimethylbenzene, to about ten days, e.g. for benzene (Atkinson and Arey, 2003).

*L46-50: There is a rich literature on many aspects of this chemistry which should be cited.*

Reply:
We agree with the referee and we added the references to the review papers by Atkinson and Arey (2003) and Vereecken (2019). For the SOA formation from aromatics oxidation we now refer to Henze et al. (2008) and Lin et al. (2012).

*L78: How were they added? i.e. what cross-sections and quantum yields used?*

Reply:

We apologize for the lack of detail here. For the photolytic HONO-formation from nitrophenols the cross sections and quantum yield provided by Chen et al. (2011) are used by the JVAL and JVPP models (Sander et al., 2014) for calculating the $j$-values. We have modified the manuscript accordingly.

In JVAL (Sander et al., 2014) the cross sections for 2-nitrophenol and 3-methyl-2-nitrophenol and the quantum yield for 2-nitrophenol by Chen et al. (2011) are used to calculate the $j$-values.

>    *L89: Please be quantitative.*

Reply:

We have added to the revised manuscript the information on the yields of glyoxal (60%) and methylglyoxal (40%) for toluene from Birdsall et al. (2010). We also specify now that these yields are for the non-radical terminating channels in the reactions with NO and $HO_2$.

We have replaced the text

Bicyclic peroxy radicals in the oxidation mechanism of toluene produce some glyoxal and methyl glyoxal as suggested by Birdsall et al. (2010). Benzene is treated analogously.

with

Bicyclic peroxy radicals in the oxidation mechanism of toluene yield 60% glyoxal and 40% methyl glyoxal from the non-radical terminating reactions with NO and $HO_2$ as suggested by Birdsall et al. (2010). Benzene is treated analogously but yields 100% glyoxal from the above mentioned reactions.

>    *L110: Please plot the data in nmol/mol to make things clearer for the reader.*

Reply:

Yes, we now plot the data with mol/mol and the appropriate exponent for the range of values shown. We agree it was not clear before.

>    *L118: Insert "surface" between these two words.*

Reply:
Done.

>    *L131-132: Can you be more specific on both the impact on the OH NH:SH ratio change and the impact on the methane lifetime.*

Reply:

Referee #1 had a similar comment and we acknowledge that the quantification of the impact on methane lifetime could have been given more space than a short mention without referring to Table 4. Therefore, we have extended the paragraph L129-132 by pointing explicitly to Table 4 and shortly discussing the changes in OH and CH4 lifetime in the two hemispheres.

The paragraph

Figure 5 shows the annual zonal mean changes of the OH mixing ratio. The changes are most pronounced in the NH upper troposphere where reductions range from 7 % to 20 %. This helps bringing

the model-simulated inter-hemispheric OH asymmetry closer to that derived from observations (**?**). Globally, aromatics oxidation reduces OH by 7.7 % and consequently increases methane lifetime.

has been extended:

Figure 5 shows the annual zonal mean changes of the OH mixing ratio. The changes are most pronounced in the NH upper troposphere where reductions range from 7 % to 20 %. These predicted changes are associated to similar reductions in $NO_x$. In fact, the upper troposphere is in general $NO_x$-limited and the oxidation of aromatics enhances the formation $N_2O_5$ and $HNO_3$ which are lost heterogeneously. This leads to an effective removal of $NO_x$ from the gas phase and lowers the radical production. The change in hemispheric burdens of OH are consistent with this picture (Table 4). This moderately helps bringing the model-simulated inter-hemispheric OH asymmetry closer to that derived from observations (**?**). Globally, aromatics oxidation reduces OH by 7.7 % and consequently increases methane lifetime by about 5.5 %. The changes are more pronounced in the northern hemisphere where aromatics are mostly emitted (Table 4). However, the latter in the EMAC model remains significantly lower than the ACCMIP multi-model mean and the observational-based estimates (**?**). Coarse model spatial resolutions (about 200 km) are known to result in an overestimation (underestimation) of global mean OH (methane lifetime) of at least 5 % (**?**). This is due to a less efficient conversion of $NO_x$ to $NO_y$ when strong pollutant emissions are artificially diluted in the model grid boxes. This aspect certainly has a larger impact on the inter-hemispheric OH asymmetry in atmospheric models that is in contrast to observational estimates (**?**).

> *L135-136: Has there been an increase in the flux through O3+OH? I'm surprised given the OH has gone down in these regions.*

Reply:
We thank the referee for spotting this inconsistency. Clearly, the simulation results do not support the statement on an increase in the flux of the $O_3$ + OH reaction in ozone-depleting regimes, e.g. over the ocean. We have removed OH from this explanation.

> *L141: Odd to ref. Fig 9 before 7 or 8. Re-order?*

Reply:
Thank you for spotting this. The figures are reordered now.

> *L153: Can you confirm which definition you used in the analysis?*

Reply:
We put this text in the caption of Table 4.

Tropospheric burdens were reckoned using six different tropopause definitions (provided by the TROPOP submodel, see Jöckel et al. (2010) for details): 1,2) surfaces of $O_3$ mixing ratio of 125 and 150 nmol/mol, respectively, 3) WMO definition (WMO (1957)), 4) dynamic PV-based (3.5 PVU potential vorticity surface, sought within 50–800 hPa), 5) climatological (invariable zonal profile, i.e. $300\text{-}215 \times (\cos(\text{latitude}))^2$ hPa) and 6) the combined definition (WMO tropopause within 30°N–30°S, otherwise dynamic PV-based tropopause). The latter definition is used by default in EMAC and in this manuscript to report tropospheric budgets. Estimated changes to tropospheric $O_3$ burden are identical within 0.05% between the available definitions, which we conclude as robust against the definition used.

> *L155: Is it widely acknowledged that it is "only" ozonolysis? And does the definition of loss change with different constructs of the O3 budget (c.f. Bates and Jacob 2019)?*

Reply:
According to the expanded definition of the odd oxygen budget by Bates and Jacob (2020), the loss ozone from reaction with phenoxy radical would count as half since $RO_2$ formation is counted with the "stoichiometric" coefficient 0.5 in the $O_y$ family. This coefficient is justified in order to account for the effect the $O(^1D) + H_2O$ reaction has on OH. However, the rationale and validity of this "stoichiometric" accounting for peroxy radicals is not clear. We agree that when ozone reacts with phenoxy radical close to the pollution sources the $NO$-to-$NO_2$ conversion by the resulting phenyl peroxy radical would largely compensate the ozone loss in question. However, in the upper troposphere where benzene is transported and where NO levels are usually low, the loss of ozone with phenoxy is a net loss.

*L156: Perhaps add e.g., as this is just one models calculation.*

Reply:
Done.

*L161: Confirm if you mean ozone or odd-oxygen?*

Reply:
We mean ozone.

*L166-167: Can you quantify the relative contribution of these different pathways to the 200 Tg/yr O3 (odd oxygen?) loss?*

Reply:
We now realize that our formulations have been not clear and misleading. The 200-300 Tg/yr we give in the manuscript is the direct ozone loss in the reaction with (substituted) phenoxy radicals. What we wanted to express here was that the phenyl peroxy radical produced by reaction R3 enhances the $NO_3$ formation at night, which in turn enhances the $O_x$ via the heterogeneous loss of $N_2O_5$. Having no detailed passive tracers for computing the $O_x$ budget in this study, we cannot quantify the strength of the $O_x$ destruction we describe. We now make this clearer in the revised manuscript.

*L174: But the way you have written R1 suggests that phenoxy radicals are not formed (instead butenedial is formed).*

Reply:
Indeed R1 destroys the aromaticity of the molecule and therefore any possibility to form further (substituted) phenoxy radicals. Unfortunately, we have not explicitly mentioned that in MCM (AROM and onlyMCM simulations) the reactions of the simplest nitrophenol (HOC6H4NO2) yield a nitrophenoxy radical which is assumed to react with $O_3$ and $NO_2$ like phenoxy radical (C6H5O). We have made this point clearer in the revised manuscript.

*Table 3: Can you confirm that these are area weighted? The surface ozone seems a bit high compared to other models I've seen.*

Reply:
Yes, they are. We have added this information in the table caption. We share the impression of the reviewer that the model computes high levels of surface ozone. We are addressing the general overestimation of tropospheric ozone by, among others, improvements of the dry deposition scheme lacking the non-stomatal sink (Emmerichs et al., 2020) and the explicit modelling of the ozone sink in cloud droplets (Rosanka et al., 2020) and deliquescent aerosols.

[revised manuscript text omitted]